# Mitonuclear genetic patterns of divergence in the marbled crab, *Pachygrapsus marmoratus* (Fabricius, 1787) along the Turkish seas

**Cansu Çetin**[1,2,3]*, **Andrzej Furman**[1], **Evrim Kalkan**[4‡], **Raşit Bilgin**[1‡]

**1** Institute of Environmental Sciences, Boğaziçi University, Bebek, Istanbul, Turkey, **2** Institute of Integrative Biology (IBZ), ETH Zürich, Zürich, Switzerland, **3** Department of Aquatic Ecology, Swiss Federal Institute of Aquatic Science and Technology, Dübendorf, Switzerland, **4** Institute of Marine Sciences, Middle East Technical University, Erdemli-Mersin, Turkey

‡ EK and RB share first authorship on this work.
* cansu.cetin@usys.ethz.ch

## Abstract

Biogeographical transition zones present good opportunities for studying the effect of the past ice ages on genetic structure of species because secondary contact zones of post-glacial lineages can be formed. In this study, we investigated the population genetic structure of the marbled rock crab, *Pachygrapsus marmoratus* along the coasts of Turkey. We genotyped 334 individuals from the Black Sea, the Turkish Straits System (TSS), the Aegean, and the Eastern Mediterranean basins. In order to reveal its evolutionary history and its population connectivity, we used mitochondrial CO1 region and five microsatellite loci. CO1 analyzes also included 610 additional samples from Genbank, which covered most of its distribution range. Both microsatellites and mtDNA showed decreased diversity in sampling sites of the TSS and the Black Sea as compared to those along the Aegean and the Levantine coasts. There is an especially strong geographical pattern in distributions of haplotypes in mtDNA, most probably as a result of genetic drift in the Black Sea and the Sea of Marmara (SoM). Microsatellite data analyses revealed two genetically distinct clusters of *P. marmoratus* (clusters *C* and *M*). While individuals belonging to cluster *C* are present in all the sampling locations, those belonging to cluster *M* are only detected along the Mediterranean coasts including the Aegean and the Levantine basins. These clusters shared similar haplotypes in the Mediterranean. Haplotypes of two sympatric clusters could be similar due to incomplete lineage sorting of ancestral polymorphisms. In order to retrieve the complex demographic history and to investigate evolutionary processes resulting in sympatric clusters in the Aegean Sea and the Levantine basin, mitochondrial markers with faster mutation rates than CO1 and/or SNP data will be useful.

## Introduction

Transition zones between different biogeographical regions are good candidates for studying how molecular markers with different resolution and effective population sizes [1] reflect the

They will be made publicly available on April 2022 or when the manuscript is published (whichever comes first). Microsatellite genotyping data is added as supplementary information. At this time, please ensure the data has been made publicly available.

**Funding:** This study was supported by a grant from the Boğaziçi University Research Fund (No: 11Y00P2) to RB. The funders had no role in study design, data collection and analysis, decision to publish, or preparation of the manuscript.

**Competing interests:** The authors have declared that no competing interests exist.

evolutionary history of the species [1]. The signature of past ice ages on genetic structure is especially apparent in biogeographical transition zones [2] where expanding post-glacial lineages tend to form hybrid zones for many terrestrial and marine species [3, 4]. Some of the examples where severe palaeogeographic or palaeoclimatic changes through geological timescales impacted population genetic structuring or mixing within species in the marine realm are the Almeria-Oran oceanographic front to the east of the Atlantic-Mediterranean interface [5], the Siculo-Tunisian Strait dividing western and eastern Mediterranean [6], North Sea-Baltic Sea transition [7, 8] and the Turkish Straits System (TSS) between the Aegean and the Black seas [9, 10].

These transition zones are often regions where divergent lineages meet in secondary contact [2]. Such secondary contact has been observed in species with very different life histories and taxonomic groups. Some examples include blue mussel (*Mytilus edulis/M. trossulus*) [11], Baltic clam (*Limecola balthica*) [8], vase tunicate (*Ciona intestinalis*) [12], European flounder (*Platichthys flesus*) [13], European sea bass (*Dicentrarchus labrax*) [14] among many others (reviewed in [15]). Despite the presence of several studies investigating the role of other transition zones in the Mediterranean in gene flow among locations [5, 16, 17], studies on the TSS are limited. The TSS (Fig 1) is a transition zone between the Black Sea and the Mediterranean Sea by means of two relatively narrow straits (the Bosphorus Strait and the Dardanelles) and an almost entirely landlocked sea (the Sea of Marmara, SoM). The Black Sea has been

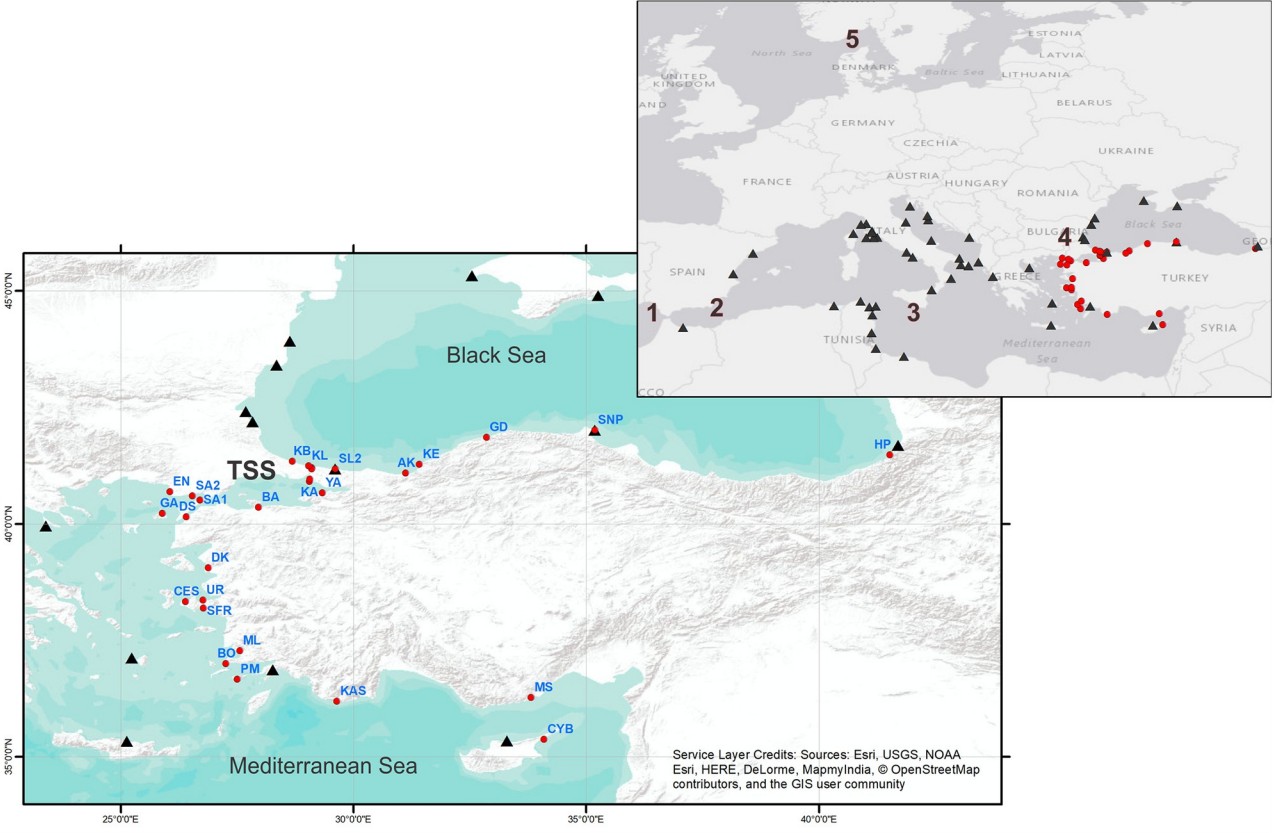

**Fig 1. Sampling locations of *Pachygrapsus marmoratus*.** Locations which were previously analyzed on mtDNA [30] and those analyzed in this study are indicated by black triangles and red circles, respectively. 1:Gibraltar Strait; 2:The Almeria-Oran oceanographic front; 3:The Siculo-Tunisian Strait; 4: The Turkish Straits System (TSS); 5:North Sea-Baltic Sea transition. Details on sampling sites of this study are indicated in S1 Table.

repeatedly connected and separated from the Mediterranean Sea through the opening and closure of the Dardanelles and Bosphorus Strait during the Quaternary period climatic fluctuations [18–20]. These fluctuations have a role in shaping the genetic diversity and diversification in and around the system [21–23]. In the Black Sea, for example, populations of various taxa have lower genetic diversity in comparison to those in the Eastern Mediterranean, due to founder effects and/or genetic drift in line with their more recent establishment [24–26]. The most recent connection between the Mediterranean and the Black Sea was established around 8,400 BP, with an inflow of its waters to the Black Sea, and TSS took its current form in terms of its hydrology and current systems around 7,500 BP [18, 19]. The current hydrology of the TSS is characterized by strong stratification (halocline and thermocline) and a two-layer current system, in which brackish water from the Black Sea flows to the Aegean above the denser saline Mediterranean waters that flow towards the Black Sea [27–29].

Episodes of Quaternary period oceanic fluctuations around the TSS and its two-layered current system makes it an interesting region wherein to study the evolutionary history of species. While the direction of the upper layer current increases one-directional transport of larvae (from the Black Sea to the Aegean), it can also limit the transport in the opposite direction as found out in studies of some marine species, such as anchovy (*Engraulis encrasicolus*) [9, 31], the Mediterranean horse mackerel (*Trachurus mediterraneus*) [32], mullets (*Liza aurata*, *L. saliens* and *Mugil cephalus*) [33], Atlantic bonito (*Sarda sarda*) [34, 35], one mammal, *Phocoena phocoena* (Linnaeus 1758) [36, 37], rock-pool prawn (*Palaemon elegans* Rathke 1837) [22, 38], and the marbled crab, *Pachygrapsus marmoratus* [30, 39]. Indeed, studies of Rossi et al. [33] and Turan [35] suggested that the recorded spatial distribution of two genetic types (II and III) within *P. elegans*, across the eastern Mediterranean and Black Sea coasts of Turkey, specifically the absence of type II in the Black Sea, could be influenced by the impact of the two-layered current regime of the TSS. Additionally, recently established water exchange through the TSS which took its current form around 7,500 BP (see [27, 29]) might have created secondary contact zones between previously separated lineages in the Black Sea and the Mediterranean. Despite the above-mentioned studies around the TSS, there is a need for studies with more detailed sampling, larger sample sizes, and by using markers with higher mutation rates than mtDNA, such as microsatellites or by using more markers (e.g. RAD sequencing), in order to reconstruct evolutionary history of species inhabiting this understudied transition zone.

In this study, we investigated the population genetic structure of the marbled rock crab *Pachygrapsus marmoratus* by using both microsatellites and mtDNA as a means to reconstruct its evolutionary history in the TSS. *P. marmoratus* has a distribution range in the Eastern Atlantic, Canary Islands, the Mediterranean and the Black Sea and it is an introduced species in the United Kingdom of Great Britain and Northern Ireland (GBIF) [40]. It is a good model species that enables investigation of the potential roles played by contemporary and historical factors in shaping the population genetic structure of marine invertebrates in the area. First of all, it is a locally abundant species with a continuous distribution throughout Mediterranean Sea and the Black Sea. It is also able to tolerate a wide range of salinity between 15–35 practical salinity unit (psu) [41] and temperature levels [42], which allows it to inhabit different habitats in the Black Sea (brackish, with surface salinity of 17–20 psu, [20, 43]) and the Mediterranean Sea (saline, with surface salinity of 38–39 psu, [20, 44]). Both its high fecundity [45] and its extended planktonic larval period [46] might result in high genetic connectivity among populations. On the other hand, oceanographic fronts can lead to differentiation in the genetic structure of the marbled crab [17].

Previous studies have found a relatively complex local genetic structure for *Pachygrapsus marmoratus* [47–50]. In terms of mtDNA, there was no separation across Gibraltar Strait and

Siculo-Tunisian Strait [48, 50], and a weak signature of divergence between Atlantic and the Mediterranean populations [51]. The most recent study with the largest sample size (587 individuals) [30], clearly differentiated three geographic groups corresponding to Black Sea, Mediterranean Sea & Canary Islands, and Portuguese Atlantic Ocean using mtDNA CO1 region (Fig 1). Kalkan et al. [39] found weak but discernible differentiation in haplotype frequencies between the Black Sea and Mediterranean populations in the mtDNA CO1 region. These results suggest that the CO1 region in this species is more informative in studies with more extensive sampling. On the other hand, several population genetic investigations along microgeographic scales using nuclear microsatellite loci revealed local genetic heterogeneity not related to geography. Patterns of chaotic genetic patchiness [52] have been found across the Italian [47–49], Tunisian [51]), Portuguese [53] coasts and most recently within the Ligurian Sea [54]. Based on these insights, the main aims of the present study are 1) to investigate the impact of the Turkish Straits System on genetic connectivity among populations of *Pachygrapsus marmoratus* from the Eastern Mediterranean and Black Sea 2) to unravel the historical as well as contemporary drivers of population genetic structuring and 3) to understand their involvement in shaping spatial distribution of genetic diversity across the TSS biogeographic transition zone using both mitochondrial and nuclear markers.

## Methods

### Sampling and DNA extraction

A total of 360 specimens of *P. marmoratus* were collected throughout its distribution range along the Eastern Mediterranean and Black Sea coasts of Turkey. Crabs were sampled from 32 sampling sites (Fig 1) between the years 2010–2014 at a depth range of 0–7 m. The geographic coordinates of the surveyed sampling sites are provided in S1 Table. After collection, crab samples were preserved in 80–99% ethanol until further processing. DNA was extracted primarily from gill tissues, and occasionally from pereiopod muscles. Gill tissue was preferred because of higher DNA concentration obtained. Genomic DNA was extracted using either Roche High Pure PCR Template Preparation Kit (Indianapolis, USA) or Invitrogen Kit (Carlsbad, CA) following the manufacturers' protocols.

### Microsatellite genotyping

360 individuals were genotyped at five microsatellite loci (Pm79 (DQ155410), Pm99 (DQ155418), Pm101 (DQ155416), Pm108 (DQ155414), Pm187 (DQ155419)) using primers designed by Fratini et al. [55]. Four of the five loci (Pm101, Pm108, Pm187, and Pm99) were amplified with the QIAGEN Multiplex PCR Kit (Hilden, Germany) following the manufacturer's protocols. The fifth locus, Pm79, was amplified based on PCR conditions used by Fratini et al. [55], with some modifications. This included 1.2 μl of DNA, 0.5 μl of each primer (10 μM), 2 μl of $MgCl_2$ (2.5mM), 0.4 μl of dNTPs (10 mM), and 0.1 μl of Taq DNA Polymerase (5 U/μl, Thermo Scientific) in a final reaction with a volume of 20 μl. Commercial fragment sizing was done on an Applied Biosystems 3730xl DNA Analyzer at Cornell University for Pm101, Pm108, Pm187, Pm99, and at Macrogen Holland for Pm79.

Microsatellite fragments were scored manually using Peak Scanner v.1 (Applied Biosystems, USA), and subsequently allele sizes were confirmed using GeneMarker v.2.6.3 (SoftGenetics, USA). Allele sizes were binned using the Excel macro FlexiBin [56]. For the subsequent analyses, individuals with missing data in at least three loci and sampling sites with less than four individuals were removed from the data set, resulting in a final sample size of 334 individuals. Genotyping errors due to null alleles, large allele drop-out or scoring of stutter peaks that can potentially lead to deviations from Hardy–Weinberg proportions were analyzed using

MICRO-CHECKER [57]. To estimate null allele frequencies, the equation 1 from Brookfield [58] was used. The software FreeNA was used (http://www1.montpellier.inra.fr/CBGP/software/FreeNA/) to compute a genotype dataset corrected for null alleles, following the ENA method described in Chapuis & Estoup [59]. Pairwise $F_{ST}$ was calculated using the *pairwise. WCfst* function in *hierfstat v. 0.5–7* following Weir and Cockerham [60]. Pairwise comparisons of $F_{ST}$ between sampling locations and their significance were assessed using bootstrapping (1000) to construct 95% confidence intervals using boot.ppfst function of hierfstat. Heatmaps of pairwise $F_{ST}$ values were created using *ggplot2 v.3.3.5* [61]. Pairwise $F_{ST}$ values calculated with corrected and uncorrected data were compared. Since overall pattern did not change and as there was a slight increase in the $F_{ST}$ values for the loci (Pm79 and Pm108) that contributed the most to the pairwise values in the corrected data, uncorrected data set was used for further analyses to decrease the probability of false positives (S3 Table).

## Analysis of genetic diversity and population genetic structure

The STRUCTURE v.2.3.4 [62] analyses comprised several runs with different burn-in lengths (50000, 20000, and 10000) and MCMC repeats (50000, 20000, and 10000), with *K* values ranging from 1 to 12. Best delta K value was estimated using the method of Evanno et al. [63] as implemented in STRUCTURE HARVESTER [64]. Once the best *K* was inferred, 20 iterations were performed using this value of *K*. Mean Q values for these iterations were calculated with CLUMPAK (Cluster Markov Packager Across K, [65], which uses a Markov clustering algorithm that relies on a similarity matrix between replicate runs, as computed by the software CLUMPP (Cluster Matching and Permutation Program) [66]. Afterwards, with CLUMPAK, a DISTRUCT [67] graph was produced for the graphical display of the results. In all analyses, uncorrelated allele frequencies were assumed to allow for admixture. STRUCTURE graphs for other K values were inspected for the presence of any significant meaningful structure (S2 Fig).

For analyses of population structure and genetic diversity, 20 sampling points with at least four individuals were used out of 32 for more statistical power (S1 Table). Hardy-Weinberg equilibrium (*HWE*) and linkage disequilibrium (*LD*) were tested in Genepop package in R v.4.7.2 [68, 69] via the Markov Chain method (dememorizations = 10000, batches = 1000, iterations = 10000) for the 20 sampling sites and for each pair of loci. The number of alleles (*Na*), the effective number of alleles (*Ne*), the expected heterozygosity (*He*), observed heterozygosity (*Ho*), and allelic richness ($A_R$) were calculated using the package diveRsity v.1.9.90 of the R software v.3.5.3. In order to check if sample sizes used in this study were representative, the relationship between the number of alleles per sampling site and sample size was visualized in a graph (S1 Fig) by using the package PopGenReport (v.3.0.5) of R software (v.3.5.3). An AMOVA [70] was performed using the poppr (v.2.8.3) package of R software (v.3.5.3) for the two STRUCTURE clusters and 20 sampling sites. The genetic relationships among sampling sites were also evaluated through a principal coordinate analysis (PCoA) carried out on the distance method of [71], as implemented with the program GenAlEx v.6.5 [72, 73].

## mtDNA cytochrome oxidase subunit I (CO1) amplification and data analyses

A fragment of the mitochondrial DNA CO1 gene was amplified in 316 individuals of *P. marmoratus* by means of the universal primers (LCO1490 and HC02198), developed by Folmer et al. [74]. 126 of these 316 individuals were sampled and analyzed previously by Kalkan et al. [39]. The PCR reaction mix contained 2.5 μl of 10X high fidelity buffer, 2.5 μl of $MgCl_2$ (25 mM), 0.5 μl of dNTPs (10 mM), 0.3 μl of each primer (20 μM), 0.25 units of Taq DNA

polymerase (5U/μl, Thermo Scientific), 2 μl of DNA (approximately 50 ng/μl) and 18.45 μl of $H_2O$ for a final reaction volume of 25 μl. PCR was performed according to the conditions described by Folmer et al. [74] with some modifications: 10 min at 94°C for initial denaturation followed by 35 cycles of 45s at 94°C for denaturation, 1 min at 50°C for annealing, and 1 min at 72°C for extension, followed by a final extension of 10 min at 72°C.

For the network analysis, 316 sequences from this study were combined with 74 CO1 haplotype sequences from GenBank (with accession numbers JF930650.1-JF930682.1 [48], KX529672.1-KX529696.1 [30], KX549320.1-KX549335.1 [51], corresponding to 610 individuals [30]). A total data set of 926 individuals were used in the network analysis (Fig 2). Sequences were edited manually with Sequencer 5.2.4 (2013 Gene Codes Corporation, Ann Arbor, USA), and aligned with ClustalX [75]. The length of sequences used in the final analyses was 490 bp. They were shorter than those of previous studies (e.g. [30] because of the different

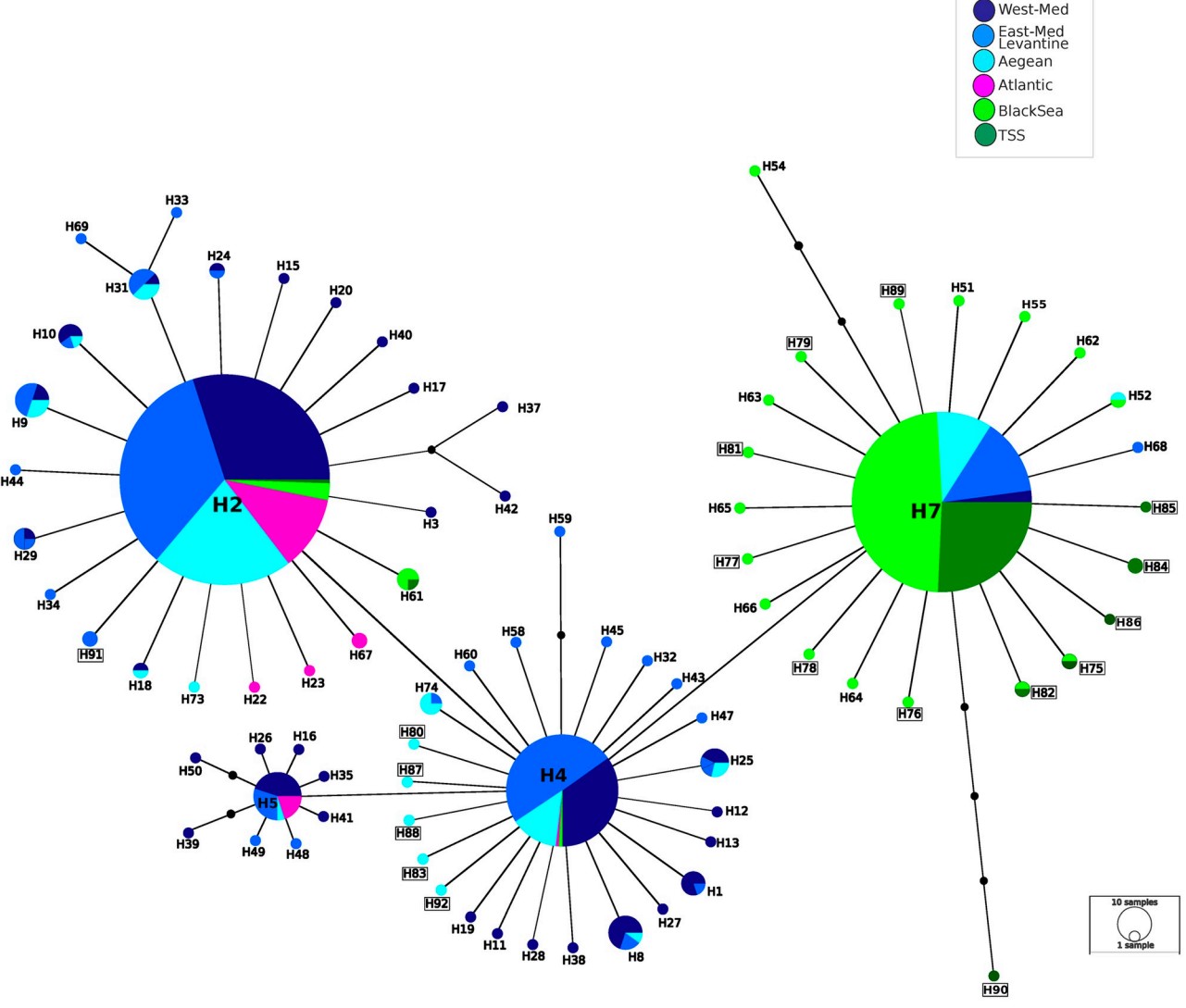

**Fig 2. Haplotype network of mtDNA CO1 sequences of *P. marmoratus*.** Network includes samples used in this study and in Kalkan et al. [39] together with GenBank sequences corresponding to 926 individuals in total. Each line corresponds to one base-pair difference and the diameter of each circle is proportional to the frequency of that haplotype. Squared names correspond to the haplotypes identified for the first time in this study.

forward primer used in our study. A haplotype network was constructed with the software Network v.4.6.1.2 [76], using the median-joining algorithm.

The same 20 sampling sites which were included in microsatellite analyses were used for analyses of genetic structure and genetic diversity in mtDNA. For each of the 20 sampling sites, haplotype diversity ($h$) and nucleotide diversity ($\pi$) were calculated with DnaSP v5.0 [77]. For the pairwise $F_{ST}$ computation and AMOVA, ARLEQUIN was used. Deviations from neutral molecular evolution were also tested with ARLEQUIN v.3.5, by calculating Tajima's $D$ [78] and Fu's $Fs$ [79, 80]. Significant negative values of these tests may suggest population expansion. $R_2$ test [81] was applied using DnaSP v.5.0 [77] to detect potential population expansion. Statistical significance was tested with coalescent simulations based on 5000 simulated resampling replicates.

## Results

### Analyses with nuclear microsatellite data

Samples from 334 individuals from 20 locations were analyzed at five nuclear microsatellite loci. After Bonferroni corrections, no linkage disequilibrium was observed for any pairs of loci, neither across clusters nor in sampling sites. There were relatively high percentages of null alleles in general. The loci Pm99 and Pm108 showed especially high percentages of null alleles in the Mediterranean (between 8–34%) (S2 Table). The locus with the least percentage of null alleles was Pm79, which also had the second highest contribution to overall $F_{ST}$ values. Genotypic errors due to large-allele drop-out were not observed in any sampling site. After comparing a subset of data with no null alleles with the whole data, we did not see any differences between population assignment values of individuals in STRUCTURE. Additionally, as mentioned in the methods, $F_{ST}$ values after the ENA correction method of FreeNA software were similar to those uncorrected (S3 Table), and thus, we did not correct for null alleles. The relationship between number of alleles and sample size suggested that number alleles tend to stabilize at around 15 individuals (S1 Fig).

The most likely number of clusters was evaluated with the ΔK method and results supported two clusters (S3 Fig). One cluster was widely distributed in all of the sampling locations (referred to as cluster $C$), and the other was restricted to the Aegean and the Levantine coasts (cluster $M$). Two clusters ($C$ and $M$) coexisted in sympatry across all sampling locations in the Aegean and the Levantine coasts of Turkey (Fig 3). Barplots for other K values (K: 3–5) were visually inspected (S2 Fig). At K = 3, some individuals belonging to Cluster $C$ formed the third cluster but this did not correspond to any geographical pattern (S2 Fig). The PCoA also reflected the differences between clusters $C$ and $M$ (Fig 4), supporting the results of the STRUCTURE analysis. The individuals that comprise the cluster $C$ were more similar to each other in terms of their genetic distances, when compared to the cluster $M$ (Fig 4). There was no pattern according to sampling sites in PCoA (S5 Fig). Among the five microsatellite loci, three (Pm108, Pm99, and Pm79) differed the most between the two clusters in terms of their allele frequency distributions (Fig 5). Pm99 and Pm101 were the most polymorphic loci, with 21 and 17 alleles, respectively. Pm79 had three alleles in the Cluster $C$, but it was dimorphic in $M$ (Fig 5). Additionally, Cluster $C$ had a dominant allele for the locus Pm79 whereas $M$ had almost equal frequencies of its alleles. Cluster $C$ was almost fixed for the locus Pm108 (Fig 5). Cluster $M$ had higher $H_e$ values than cluster $C$, 0.60 versus 0.50, respectively (S4 Table). Individuals belonging to cluster $C$ in the Mediterranean ($C2$) had higher $H_e$ values (0.58), when compared to those in the SoM and the Black Sea ($C1$) (0.45) (S4 Table).

Pairwise $F_{ST}$ tests between pairs of sites confirmed that sampling sites were genetically different (global $F_{ST}$: 0.11, p < 0.05; Fig 6). There was a significant differentiation between

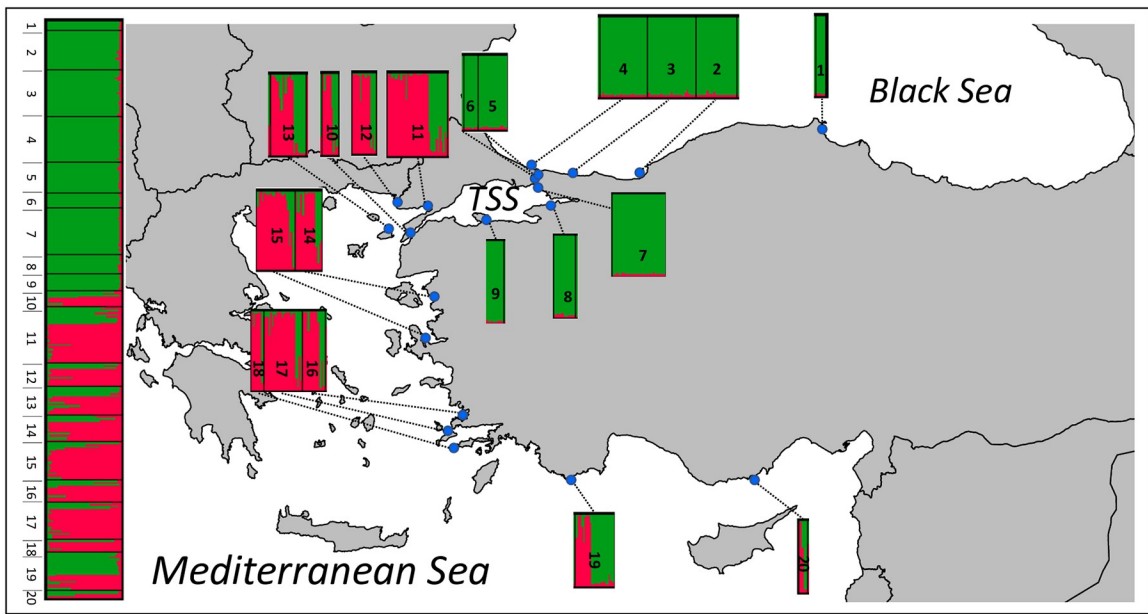

**Fig 3. STRUCTURE results based on five microsatellite loci at K = 2 (on the left) and a map showing the geographical distribution with *P. marmoratus* sampling sites (on the right).** Each vertical bar of the STRUCTURE plot represents an individual (y axis) with the colors indicating their associated probability of belonging to one of the two genetic clusters (x axis). Black lines separate the 20 sampling sites. The clusters *M* and *C* are shown in red and green, respectively.

sampling sites in the Black Sea, the Bosphorus Strait & the SoM (1–9) and those in the Dardanelles & the Mediterranean (10–20) (Fig 6). Although sample size of Dardanelles (N = 10) is too small to make strong inferences, it should be noted that Dardanelles samples (sampling site 10), although considered as part of the TSS, were differentiated from the other sampling sites in the TSS and the Black Sea. Furthermore, $F_{ST}$ values (Fig 6) did not support its differentiation from the samples in the Mediterranean. Sampling sites in the Black Sea, the Bosphorus Strait and in the SoM were not differentiated from each other except very low levels of differentiation between sampling sites 4 vs 9 and 4 vs 1 (Fig 6). Sampling site 19 was significantly differentiated from some of the sampling sites (sites 10, 11 and 18) in the Mediterranean with no geographical pattern (Fig 6).

An AMOVA was performed by using the microsatellite data set. We tested the effect of geography on the distribution of genetic variation in microsatellites by grouping populations (sampling sites) into three geographic groups corresponding to the Black Sea, the TSS and the Mediterranean Sea (Table 1). The largest percentage of variance was explained by variation within sampling sites (81%), followed by variation between three geographic groups (16%) (Table 1).

## Main patterns in mtDNA CO1 data

A haplotype network based on 490 bp fragments of the mtDNA CO1 region, included 926 *P. marmoratus* samples from the Mediterranean, the Black Sea, and the Atlantic Ocean (Fig 2). Overall, 80 haplotypes were observed. Since the fragment size in this study was smaller than used in Fratini et al. [30], some of the previously separate haplotypes were grouped together. This decreased the number of 74 haplotypes reported previously in Fratini et al. [30, 48] to 62. Haplotypes H1-H74 corresponded to haplotypes previously reported in the studies of Fratini et al. [30, 48]. Among the samples of our study, there were in total 31 haplotypes, 18 of which

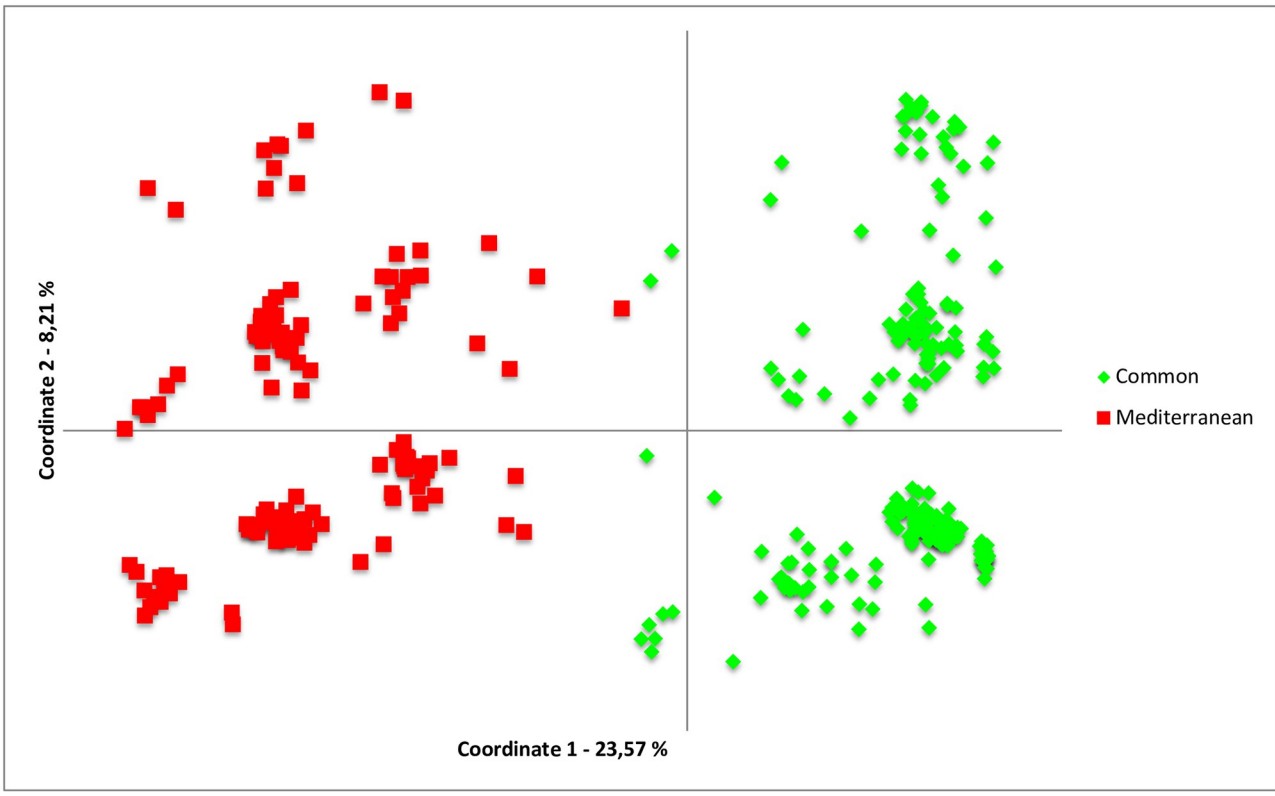

**Fig 4. PCoA based on five microsatellite loci for *P. marmoratus* individuals.** Colors denote two STRUCTURE clusters: Mediterranean (red square) and Common (green diamonds).

were recorded for the first time (haplotypes H75-H92). The network showed a star-like pattern with only two bp separating the three most common haplotypes (H2, H4 and H7), corresponding to 0.4% divergence. The most common haplotype, H2 has a wide geographic distribution covering most of the range of *P. marmoratus*. It is found in 384 individuals (corresponding to 41% of all samples). Around 52% (328 individuals) of the samples from the Mediterranean Sea, 83% (44 individuals) from the Atlantic Ocean, 6% (10 individuals) from the Black Sea, and 1% (2 individuals) from the TSS have this haplotype. The distribution of the second most common haplotype, H7, is the dominant haplotype in the Black Sea, corresponding to 81% of Black Sea (136 individuals), 87% of the TSS (72 individuals), and 12% of the Mediterranean (73 individuals) samples.

Sampling sites inhabiting the Mediterranean were genetically more diverse than those in the Black Sea and the Sea of Marmara (Table 2). Among the 31 haplotypes included in the dataset sampled in this study, there were 64 polymorphic sites with a total of 67 mutations. The average haplotype and nucleotide diversities were 0.72 (SD = 0.01) and 0.0027 (SD = 0.00006), respectively. Most of the mutations involved synonymous changes to amino acid positions, whereas 10 of the 67 mutations corresponded to non-synonymous sites in the coding region. Nucleotide diversity values in all of our sampling sites were low ($< 0.0035$); with the ones in the Black Sea and in the SoM having lower values (0–0.0015) than those in the Mediterranean (0.0014–0.0031). Haplotype diversities ranged from 0.00 to 0.810 (Table 2). The highest nucleotide and haplotype diversities were recorded in the sampling sites in the Aegean (see sampling sites 10, 13, and 17 in Table 2).

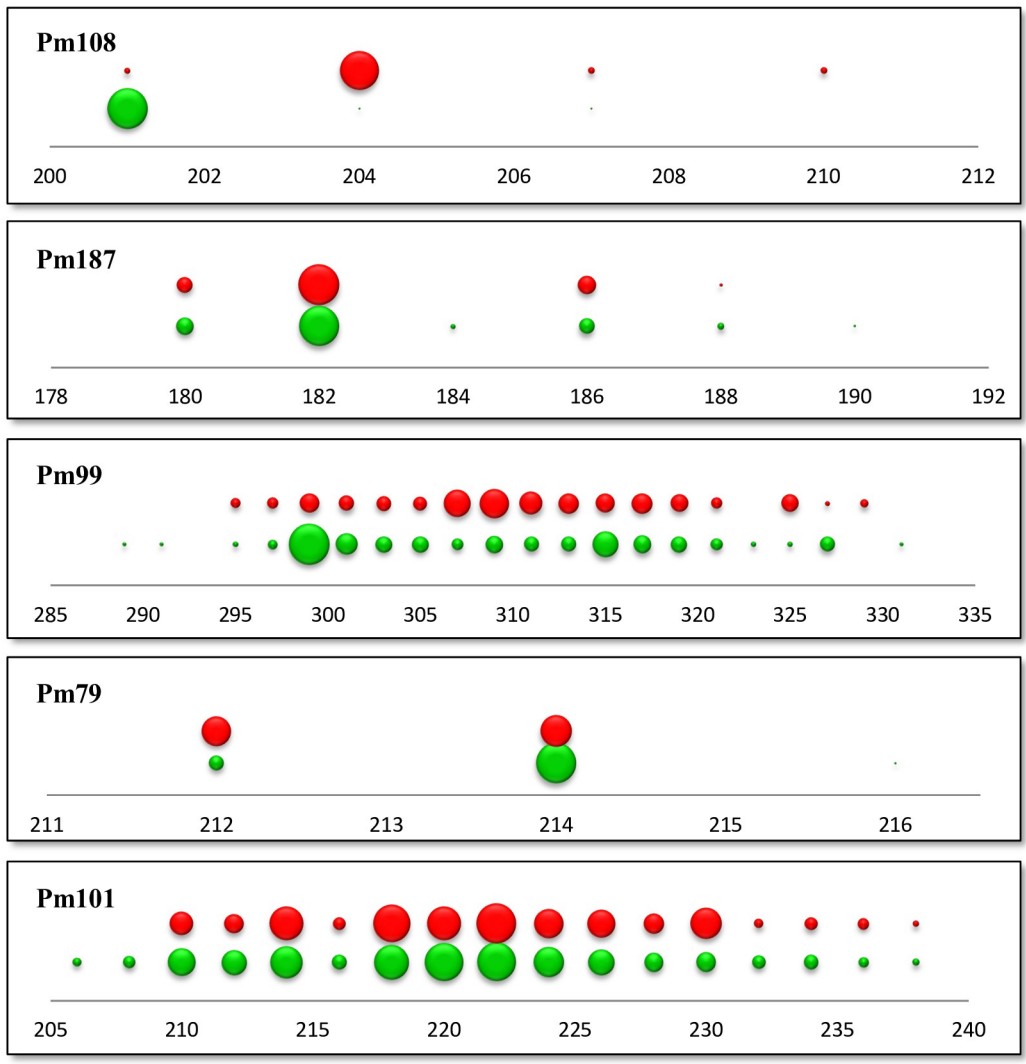

**Fig 5. Allele frequency distributions of five microsatellite loci.** In each panel, the upper row belongs to the cluster *M* and the lower row belongs to the cluster *C*. Numbers in the x-axis correspond to base pairs. The diameter of each circle is proportional to the number of individuals with the corresponding allele.

Two different AMOVAs were tested by using the mtDNA dataset. First, we tested the effect of geography on the distribution of genetic variation by grouping populations (sampling sites) into three geographic groups (Table 3). Then, we tested the effect of geography by grouping populations (sampling sites) into three geographic groups corresponding to the Black Sea, the TSS and the Mediterranean Sea (Table 3). In both cases, the largest percentage of variance was explained by variation within sampling sites (52% vs. 48%, respectively for grouping based on the geography and on the clusters), followed by variation between three geographic groups (47%) and the two clusters (36%) (Table 3). When we grouped sampling sites based on geography, $F_{CT}$ value was recorded as 0.47 ($P < 0.001$) (Table 3). Secondly, grouping according to microsatellite clusters resulted in a lower $F_{CT}$ value than the grouping based on geography (0.39, $P < 0.001$ vs. 0.47, $P < 0.001$, respectively) (Table 3). Among the 316 sequenced individuals, only 111 specimens assigned to cluster *M*, with a total of 15 haplotypes and 13

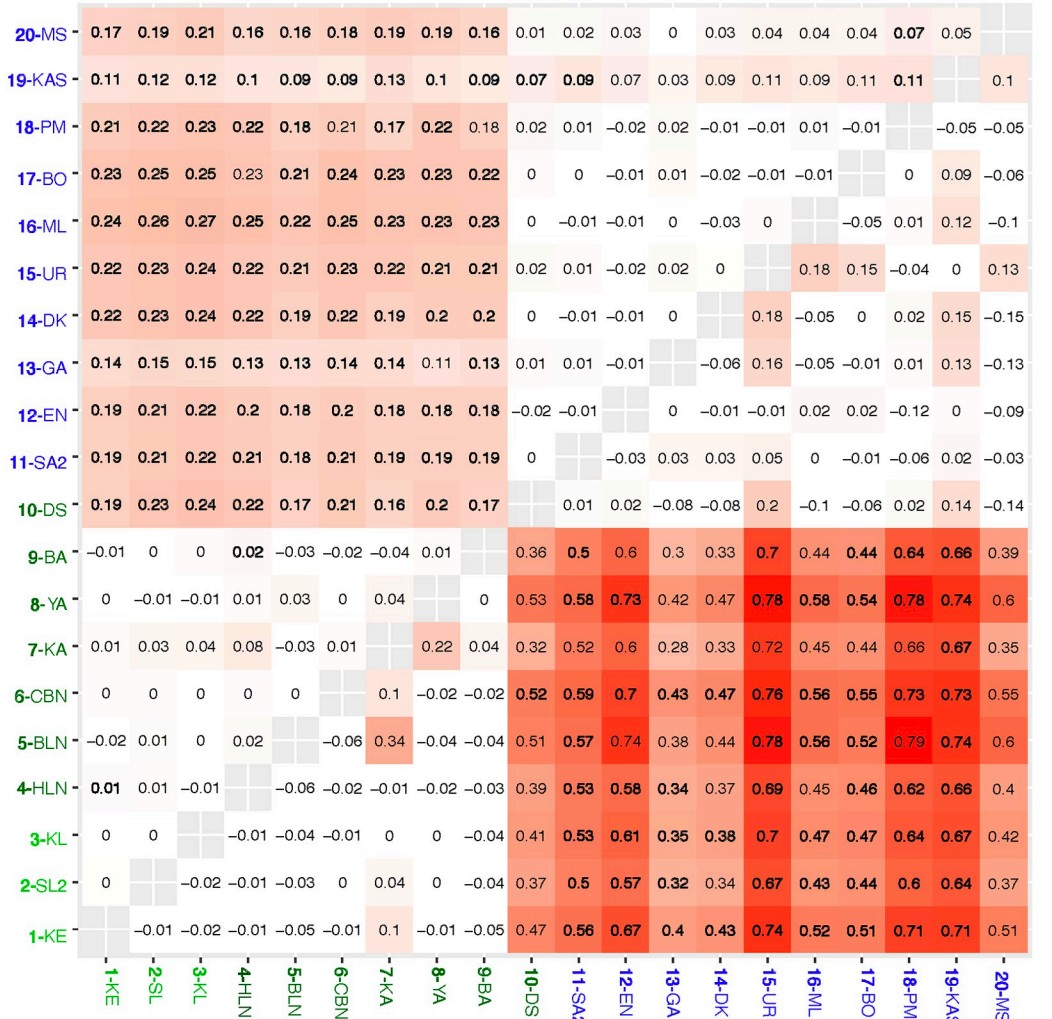

**Fig 6. The pairwise $F_{ST}$ values of microsatellite loci (above diagonal) and of mitochondrial genetic divergence (below diagonal).** Abbreviations of 20 sampling sites are denoted with their sampling codes. Colors on the fonts correspond to three geographical regions: Blue: Mediterranean; Dark green: TSS; Green: Black Sea. were estimated from sequence divergence data. Significant values are in bold (significance of microsatellites based on 95% confidence intervals and of mtDNA based on P value of <0.0003 after Bonferroni correction).

**Table 1. Analysis of molecular variance (AMOVA) table on 20 sampling sites and across three groups according to geography and 20 sampling sites using five microsatellite loci.**

| Source of variation | SS | Variance components | Variance (%) | F statistics | P value |
|---|---|---|---|---|---|
| Among groups | 111.48 | 0.49 | 15.61 | $F_{GT} = 0.19$ | < **0.001** |
| Among sampling sites/groups | 71.68 | 0.10 | 3.29 | $F_{PC} = 0.04$ | < **0.001** |
| Within sampling sites | 802.53 | 2.55 | 81.10 | | |
| Total | 985.70 | 3.15 | | | |

Significant P values are in bold (1000 permutations). Three groups are the Black Sea, the TSS, and the Mediterranean. The second column refers to sum-of-squares (SS).

**Table 2. Genetic diversity and demographic parameters in *Pachygrapsus marmoratus* clusters and sampling sites based on CO1 region.**

| Code | Sampling Site | Basin | N | Nh | Np | h | π | D | F$_S$ | R$_2$ |
|------|---------------|-------|---|----|----|---|---|---|-------|-------|
| 1 | KE | Black Sea | 25 | 4 | 4 | 0.230 (0.11) | 0.0008 (0.0000) | | | |
| 2 | SL2 | | 28 | 5 | 5 | 0.381 (0.12) | 0.0013 (0.0004) | | | |
| 3 | KL | | 26 | 5 | 5 | 0.351 (0.12) | 0.0011 (0.0004) | | | |
| 4 | HLN | | 20 | 4 | 7 | 0.284 (0.13) | 0.0014 (0.0008) | | | |
| 5 | BLN | Turkish Straits System | 8 | 1 | 0 | 0.000 (0.00) | 0.0000 (0.0000) | | | |
| 6 | CBN | | 27 | 5 | 5 | 0.279 (0.11) | 0.0008 (0.0004) | | | |
| 7 | KA | | 3 | 2 | 1 | 0.667 (0.31) | 0.0014 (0.0006) | | | |
| 8 | YA | | 12 | 2 | 1 | 0.167 (0.13) | 0.0003 (0.0003) | | | |
| 9 | BA | | 12 | 2 | 3 | 0.167 (0.13) | 0.0010 (0.0008) | | | |
| 10 | DS | | 7 | 4 | 3 | 0.810 (0.13) | 0.0027 (0.0005) | | | |
| 11 | SA2 | Mediterranean/Northern Aegean | 33 | 8 | 6 | 0.646 (0.08) | 0.0021 (0.0004) | | | |
| 12 | EN | | 10 | 2 | 2 | 0.467 (0.13) | 0.0019 (0.0005) | | | |
| 13 | GA | | 14 | 5 | 5 | 0.725 (0.09) | 0.0030 (0.0004) | | | |
| 14 | DK | Mediterranean/ Southern Aegean | 13 | 4 | 3 | 0.654 (0.11) | 0.0025 (0.0004) | | | |
| 15 | UR | | 18 | 5 | 5 | 0.405 (0.14) | 0.0013 (0.0005) | | | |
| 16 | ML | | 13 | 3 | 2 | 0.692 (0.08) | 0.0019 (0.0003) | | | |
| 17 | BO | | 19 | 6 | 5 | 0.743 (0.07) | 0.0023 (0.0004) | | | |
| 18 | PM | Mediterranean /Levantine | 8 | 3 | 2 | 0.464 (0.20) | 0.0014 (0.0006) | | | |
| 19 | KAS | | 18 | 5 | 4 | 0.614 (0.12) | 0.0016 (0.0004) | | | |
| 20 | MS | | 5 | 2 | 2 | 0.600 (0.18) | 0.0025 (0.0007) | | | |
| M | Mediterranean | | 111 | 15 | 13 | 0.611 (0.05) | 0.0020 (0.0002) | -3.3. | -10.4 | 0.04 |
| C | Common | | 206 | 20 | 22 | 0.475 (0.04) | 0.0018 (0.0002) | -2.1 | **-19.2** | **0.02** |
| C1 | Common in the north of Dardanelles | | 160 | | | | | -2.3 | **-18.8** | 0.02 |
| C2 | Common in Mediterranean | | 46 | | | | | -0.8 | -2.7 | 0.08 |

Number of individuals (*N*), number of haplotypes (*Nh*), number of polymorphic sites (*Np*), haplotype (*h*) and nucleotide (*π*) diversities per site, Tajima's *D* parameter, Fu's *F$_s$*, and *R$_2$* parameter for each sampling site and also for Common cluster (*C*), its sampling sites in the north of the Dardanelles (*C1*), sampling sites of C in the Mediterranean (*C2*), and Mediterranean (*M*) sampling sites. The significant *P* values are reported in bold.

polymorphic sites (Table 2). The remaining 206 individuals were found to belong to cluster *C*, with a total of 20 haplotypes and 22 polymorphic sites (Table 2). Haplotype diversity of the smaller cluster, *M*, was higher than the larger one, *C* (0.611 vs. 0.475, respectively). However, the nucleotide diversities of the two clusters were comparable (Table 2). The most common

**Table 3. Analysis of molecular variance (AMOVA) table using mtDNA CO1 subunit among three groups according to geography and 20 sampling sites and among two clusters and 20 sampling sites.**

| Source of variation | SS | Variance components | Variance (%) | F statistics | P value |
|---------------------|-----|---------------------|--------------|--------------|---------|
| Among groups | 68.91 | 0.34 | 46.98 | F$_{CT}$ = 0.47 | **< 0.001** |
| Among sampling sites/groups | 8.17 | 0.01 | 0.90 | F$_{SC}$ = 0.02 | NS |
| Within sampling sites | 112.95 | 0.38 | 52.12 | F$_{ST}$ = 0.48 | **< 0.001** |
| Total | 190.02 | 0.73 | | | |
| Among clusters | 41.89 | 0.28 | 36.32 | F$_{CT}$ = 0.39 | **< 0.001** |
| Among sampling sites/clusters | 44.48 | 0.12 | 15.61 | F$_{SC}$ = 0.21 | **< 0.001** |
| Within sampling sites | 104.31 | 0.37 | 48.07 | F$_{ST}$ = 0.52 | **< 0.001** |
| Total | 190.69 | 0.76 | | | |

Three groups are the Black Sea, the TSS, and the Mediterranean. Significant *P* values are in bold. The second column refers to sum-of-squares (SS).

haplotypes in cluster *C* and *M* were haplotypes H7 and H2, respectively (S6 Fig). There was significant genetic differentiation between sampling sites (global $F_{ST}$ = 0.38, p = 0.0004). The main genetic differentiation between sampling sites was between those in the Mediterranean region (sampling sites 10–20) and those in the Black Sea and the SoM (sampling sites 1–9), with $F_{ST}$ values ranging from 0.28 to 0.78 (Fig 6). The sampling site in the Dardanelles (sampling site 10) was more similar to sampling sites in the Mediterranean rather than those in the TSS (Fig 6).

In order to infer demographic history of populations, we used three different neutrality tests (the Tajima's *D*, the Fu's $F_s$, and the $R_2$ test) [82]. Negative values of the Tajima's *D* and the Fu's $F_s$ parameters together with low values of $R_2$ suggested strong signals for population expansion [80] in the cluster *M* and in the cluster *C* inhabiting the Black sea and the TSS (*C1*) (Table 2). Individuals belonging to cluster *C* in the Mediterranean (*C2*) did not show any significant signals of population expansion (Table 2).

## Discussion

Our results show that different markers, microsatellites and mtDNA, show much lower genetic diversity in the Black Sea and the SoM compared to the Mediterranean Sea. Most probably, stronger genetic drift in the Black Sea and the SoM as a result of their isolation in the Pleistocene climatic fluctuations has led to this pattern [18–20]. There is an especially strong geographical pattern in distributions of haplotypes in mtDNA between different basins (the Black Sea, the TSS, and the Mediterranean). This is not very surprising given that effective population size ($N_e$) of mtDNA is one-quarter of that nuclear genome; thus, it will sort faster when there is a genetic break between lineages [83]. Specifically, H7 is the dominant haplotype in the Black Sea and its surrounding haplotypes are only found in the Black Sea or in the TSS. This pattern implies that H7 was isolated for a long enough time in the Black Sea and the TSS to have resulted in 20 haplotypes detected in the region. One base pair difference between H7 and these haplotypes corresponds to at least around 125,000 years based on mutation rate of 1.66% per million years [84]. Additionally, surrounding haplotypes of H4 and H5 are not present in the Black Sea or in the TSS, suggesting the geographical isolation of populations from the Black Sea and the TSS.

The geographical distribution of genetic variation in microsatellites shows the signature of more contemporary gene flow between the sampling sites in the Black Sea and those in the Mediterranean. Our analyses with microsatellites reveal two distinct genetic clusters in *P. marmoratus* (cluster *C* and *M*) along Turkish coasts. Whereas cluster *C* is present in both the Black Sea, the SoM and the Mediterranean, cluster *M* is not present in the Black Sea and the SoM. The sampling sites in the Mediterranean have both clusters *C* and *M* in sympatry. Visual representation of mtDNA haplogroups combined with microsatellite clustering of individuals (S4 Fig) reveal that in the Black Sea and the TSS, individuals from cluster *C* have mostly one mitochondrial haplogroup (H7, which is the dominant haplotype in the Black Sea) whereas the most common haplogroups H2 and H7 are shared in the sympatric clusters in the Mediterranean without any pattern. Retention of ancestral polymorphisms is a plausible explanation for the pattern in CO1, supported by lack of a clear admixture zone in the Mediterranean [83].

Below, the hypothetical origins of these two clusters are discussed below by referring to the geological history of the regions of interest. Based on our scenario, cluster *M* would be the local population in the Mediterranean. On the other hand, cluster *C*, differentiating from cluster *M* in the Black Sea during the Last Glacial Maximum (LGM), would have subsequently dispersed to eastern Mediterranean after the establishment of the current system in the TSS around 7,500 BP. After the opening of the Dardanelles for the first time since the formation of

the Tethys Sea in the Riss-Wurm Interglacial (150,000–100,000 BP), ancestral *P. marmoratus* populations might have dispersed from the Mediterranean into the Black Sea, which had higher salinity than the present [20]. This hypothesis is also in line with the findings of Fratini et al. [30] who found a signature of demographic expansion for the Mediterranean metapopulation dated around 180,000–60,000 BP. During this range expansion, the expanding population had likely experienced a population bottleneck that resulted in the loss of genetic variation in the Black Sea. Especially, sampling sites in the Black Sea, the Bosphorus Strait and the SoM (1–9) had the lowest haplotype diversity values (ranging from 0–0.4). Low diversity in Black Sea sampling sites with respect to those in the Mediterranean in terms of mtDNA has also been shown in other marine species, including anchovy [9] and a chaetognath [85]. Genetic drift in the Black Sea, the Bosphorus Strait and the SoM sampling sites is also supported by higher *He* values of sampling sites in the Mediterranean (between 0.54–0.67), when compared to sampling sites in the Black Sea and the SoM populations (between 0.40–0.45) in microsatellites.

During the peak of the Last Würm Glaciation (24,000–20,000 BP), the Black Sea population of the species could have survived in isolation, and differentiated from the Mediterranean populations. Under this isolation, cluster *C* and cluster *M* would have differentiated in the Black Sea and in the Mediterranean, respectively. Similar suggestions were also made for the shrimp species *Crangon crangon* [86] and *P. elegans* [87]. Subsequently, population *C* would have expanded its range into the Mediterranean when the connection between the Black Sea and the Mediterranean was re-established. In other words, a refugial population of the ancestral *C* population might have survived in the SoM, which probably had relatively higher salinities during the LGM. According to this hypothesis, two populations would be isolated approximately for only about 15,000 years, from the peak of the Last Würm Glaciation (24,000–20,000 BP) to the establishment of the current system in the TSS around 7,000 years ago. The similarity in CO1 data of sympatric clusters in the Mediterranean can be attributed to incomplete lineage sorting in mtDNA as this time span is inadequate to detect post-isolation differentiation, considering the slow mutation rate of the CO1 gene [84]. However, it should be noted that, since the detailed biogeographic and evolutionary history of *P. marmoratus* has not yet been constructed throughout its distribution range, proposed dates of origins of these two clusters is still disputable.

Both mtDNA and nuclear microsatellites showed strong differentiation between sampling sites from the Black Sea and the Mediterranean in a concordant manner, complementing the results of Fratini et al. [30]. This pattern is similar to other species such as anchovies [9], the black scorpionfish (*Scorpaena porcus*) [26], and the Mediterranean mussel (*Mytilus galloprovincialis*) [10], among others. For example, Paterno et al. [10], by using genome-wide single nucleotide polymorphisms (SNP), revealed differentiation between the Black Sea and the Mediterranean samples. They also found that Mediterranean sampling sites of *M. galloprovincialis* had mixed ancestry whereas Black Sea sampling sites were more homogeneous. This pattern is similar to more heterogeneous assignment probabilities of *P. marmoratus* individuals in the Mediterranean in our study (Fig 3). Boissin et al. [26] also revealed Black Sea-Mediterranean separation in *S. porcus* by microsatellites, but not by mtDNA. However, neither of these studies could reveal the pattern around the TSS due to a lack of samples from adjacent basins to TSS [9, 26, 88].

Larger genetic variation within sampling sites than those between them (see Table 1 and S5 Fig) suggest a heterogenic genetic pool which was also observed in previous studies across the Italian [47–49], Tunisian [51], Portuguese [53] and Ligurian coasts [54]. Additionally, K = 3 barplot of the STRUCTURE analysis showed heterogeneity of sampling sites mainly involving cluster *C* with no geographical structure (S2 Fig). Previously suggested ecological and

biological processes such as differences in reproductive success, larval dispersion pattern and local larval retention could also be responsible for heterogeneity of genetic pool in this study [54]. However, heterogeneity of STRUCTURE clusters at K 3–5 could also be an artefact of the small number of markers used. We think that using more genetic markers such as SNPs, better inferences of the drivers of genetic structure in the region could be possible with more detailed analyses such as Spatial Analysis of Molecular Variance (SAMOVA) and genetic relatedness.

Biogeographical transition zones present good opportunities for studying the effect of the past ice ages on genetic structure of terrestrial and marine species because secondary contact zones of post-glacial lineages can be formed [3,4]. In our study, by focusing on one of the less studied transition zones in the Mediterranean, the Turkish Straits System, we found divergent *P. marmoratus* clusters in sympatry in the Eastern Mediterranean basins, suggesting a secondary contact zone between previously isolated populations. This pattern was not detected by mtDNA which mostly reflected the geographic differentiation between the Black Sea and the Mediterranean. In order to retrieve the complex demographic history and to investigate evolutionary processes resulting in sympatric clusters in the Aegean Sea and the Levantine basin, mitochondrial markers with faster mutation rates than CO1 and/or SNP data will be useful.

## Supporting information

**S1 Fig. Number of alleles for each sampling location vs. sample size of sampling locations.** X axis shows the sample sizes of different sampling sites and Y axis shows the total number of alleles corresponding to each site.
(PDF)

**S2 Fig. STRUCTURE plots for K = 2, K = 3, K = 4 and K = 5.** Individuals belonging to cluster *C* are depicted in blue and those belonging to cluster *M* are depicted in orange.
(PDF)

**S3 Fig. Methods to evaluate the most likely partition of the data in STRUCTURE analysis.** a) Evanno's statistic, Delta K values as a function of K, K = 1–5, averaged over 20 runs. b) Mean log probability of data (-LnPr) of K = 1–5.
(PDF)

**S4 Fig. Visual representation of microsatellite data combined with mtDNA data.** STRUCTURE Q values were ordered in each pooled group of sampling sites (green-population C, red-population M) and are shown in the bottom row. Each bar belongs to one individual. Corresponding mtDNA haplo-groups (green: main haplotype of H7 and its derived haplotypes, red: main haplotypes of H2 and H4 and their derived haplotypes, white: missing data) are indicated on the top row.
(PDF)

**S5 Fig. PCoA based on five microsatellite loci for P. marmoratus individuals.** Colors denote original 32 original sampling sites and rectangles denote STRUCTURE clusters (green: *C*; red: *M*).
(PDF)

**S6 Fig. Geographical distribution of the percentages of three most common haplotypes in sampling sites pooled for easier representation.** Black: H7; Light gray: H2; Dark gray: H4; White: individuals with less represented haplotypes.
(PDF)

**S1 Table. Sampling sites and their corresponding coordinates.** Codes indicate the abbreviations used in the map in Fig 1. The information about the original 32 sampling sites and their sample sizes are denoted in corresponding columns. In the column called "sampling sites", those which were omitted from analyses except PCoA are denoted in gray whereas the others are denoted in black.
(XLSX)

**S2 Table. Percentages of loci with significant null alleles in 20 sampling sites.** Significant loci determined by HWE test after Bonferroni correction are in bold.
(PDF)

**S3 Table. Average $F_{ST}$ for all loci and individual contribution of loci for $F_{ST}$ values between populations.** $F_{ST}$ values calculated with FreeNA corrected genotypes and not-corrected genotypes are indicated.
(PDF)

**S4 Table. Genetic variability of the five microsatellite loci for 20 sampling sites and clusters *C* and *M* of *P. marmoratus*.** *N*, sample size; *Na*, number of alleles; RS, allelic richness; $H_o$, observed heterozygosity; $H_e$, expected heterozygosity; F, fixation index; Av, average.
(CSV)

**S5 Table. Genotypes of 351 individuals scored for five microsatellite regions.** Corresponding sampling codes, and geographical basins are included as columns.
(CSV)

## Acknowledgments

We especially thank Mert Elverici, S. Ünsal Karhan, Kübra Karaman, Gulsen Cetin, Canan Cetin, Nazan Yürür, Sena Ozbek, Ceren Cekmer, Burcu Demirag, Şebnem Ozaltın, Cevat Cetin, Beyhan Cekmer, Feriha Cetin, Mehmet Yürür, Ozer Cekmer and Aysu Cetin for their help in the fieldwork. We also thank Jukka Jokela, Mustafa Yücel, Devrim Tezcan and anonymous reviewers for their comments on the paper.

## Author Contributions

**Conceptualization:** Evrim Kalkan, Raşit Bilgin.

**Data curation:** Cansu Çetin.

**Formal analysis:** Cansu Çetin, Andrzej Furman.

**Funding acquisition:** Raşit Bilgin.

**Investigation:** Cansu Çetin, Evrim Kalkan.

**Methodology:** Andrzej Furman, Evrim Kalkan, Raşit Bilgin.

**Project administration:** Raşit Bilgin.

**Resources:** Raşit Bilgin.

**Software:** Cansu Çetin.

**Supervision:** Andrzej Furman, Evrim Kalkan, Raşit Bilgin.

**Validation:** Cansu Çetin, Andrzej Furman, Evrim Kalkan, Raşit Bilgin.

**Visualization:** Cansu Çetin, Evrim Kalkan.

**Writing – original draft:** Cansu Çetin.

**Writing – review & editing:** Andrzej Furman, Evrim Kalkan, Raşit Bilgin.

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
