## [Decision Letter · Decision Letter 0]

29 Jun 2021

PONE-D-21-13064

Genetic divergence in microsatellites but not in mtDNA in sympatric populations of the marbled crab, <pachygrapsus marmoratus=""> (Fabricius, 1787) along the Turkish seas

PLOS ONE

Dear Dr. Çetin,

Thank you for submitting your manuscript to PLOS ONE. After careful consideration, we feel that it has merit but does not fully meet PLOS ONE’s publication criteria as it currently stands. Therefore, we invite you to submit a revised version of the manuscript that addresses the points raised during the review process.</pachygrapsus>

The study represent interesting populations across a special setting. There are numerous weaknesses with the paper. The presentation of results should be improved, questions made more clear, the specific results interpreted in context of the question and the analyses need to be improved (added in some case). In some cases they have to be reinterpreted or better described. The six main points are the following, but the three reviewers also give many extra points for corrections and adjustments.

The frequency of mt haplogroups differs by areas,I would argue that the authors should show that there are no signals in higher significant K´s.Present the cytonuclear pattern (microsatellite cluster - mtDNA haplotype combination) to gain further insight into the patterns of admixture in the TSS.Perform isolation with migration (IMA) analyses to test mitochondrial gene flow between clustersGet a native English speaker to help with language, either an academic or certified English copy-editor.Supplementary figures and tables, add titles and more detailed legends and descriptions.

We look forward to receiving your revised manuscript.

Kind regards,

Arnar Palsson, Ph.D.

Academic Editor

PLOS ONE

Additional Editor Comments:

The study represent interesting populations across a special setting. There are numerous weaknesses with the paper. The presentation of results should be improved, questions made more clear, the specific results interpreted in context of the question and the analyses need to be improved (added in some case). In some cases they have to be reinterpreted or better described. The six main points are the following, but the three reviewers also give many extra points for corrections and adjustments.

1. The frequency of mt haplogroups differs by areas,

2. I would argue that the authors should show that there are no signals in higher significant K´s.

3. Present the cytonuclear pattern (microsatellite cluster - mtDNA haplotype combination) to gain further insight into the patterns of admixture in the TSS.

4. Perform isolation with migration (IMA) analyses to test mitochondrial gene flow between clusters

5. Get a native English speaker to help with language, either an academic or certified English copy-editor.

6. Supplementary figures and tables, add titles and more detailed legends and descriptions.

Journal Requirements:

4. We note that Figures 1 and 2 in your submission contain map images which may be copyrighted. All PLOS content is published under the Creative Commons Attribution License (CC BY 4.0), which means that the manuscript, images, and Supporting Information files will be freely available online, and any third party is permitted to access, download, copy, distribute, and use these materials in any way, even commercially, with proper attribution. For these reasons, we cannot publish previously copyrighted maps or satellite images created using proprietary data, such as Google software (Google Maps, Street View, and Earth). For more information, see our copyright guidelines: http://journals.plos.org/plosone/s/licenses-and-copyright.

You may seek permission from the original copyright holder of Figures 1 and 2 to publish the content specifically under the CC BY 4.0 license. 

If you are unable to obtain permission from the original copyright holder to publish these figures under the CC BY 4.0 license or if the copyright holder’s requirements are incompatible with the CC BY 4.0 license, please either i) remove the figure or ii) supply a replacement figure that complies with the CC BY 4.0 license. Please check copyright information on all replacement figures and update the figure caption with source information. If applicable, please specify in the figure caption text when a figure is similar but not identical to the original image and is therefore for illustrative purposes only.

5. Please include captions for ALL your Supporting Information files at the end of your manuscript, and update any in-text citations to match accordingly. Please see our Supporting Information guidelines for more information: http://journals.plos.org/plosone/s/supporting-information.

Reviewers' comments:

Reviewer's Responses to Questions

**Comments to the Author**

1. Is the manuscript technically sound, and do the data support the conclusions?

Reviewer #1: Partly

Reviewer #2: Partly

Reviewer #3: Partly

2. Has the statistical analysis been performed appropriately and rigorously? 

Reviewer #1: Yes

Reviewer #2: Yes

Reviewer #3: Yes

3. Have the authors made all data underlying the findings in their manuscript fully available?

Reviewer #1: Yes

Reviewer #2: No

Reviewer #3: Yes

4. Is the manuscript presented in an intelligible fashion and written in standard English?

Reviewer #1: Yes

Reviewer #2: No

Reviewer #3: Yes

5. Review Comments to the Author

Reviewer #1: Comments PLOS ONE

The authors analyze microsatellite and mtDNA data from the marbled crab in the Turkeys, Black and Mediterranean Sea. The analysis are restricted to 5 microsatellites and the CO1 region of the mitochondrial DNA. The authors find a genetic difference between the Black and Turkeys Sea on one side and the Mediterranean Sea on the other when looking at the microsatellite data, but not when analyzing the mtDNA data.

Overall, I find this study quite interesting, and a presumable need for further knowledge about the crab. It would have been nice to see more independent markers, but I also realize you sometimes work with what you have. But it would be nice to see some SNP data for these crabs in the future.

I am not sure to what extend I agree about the overall conclusion of the paper. The author states there is no genetic divergence in the mtDNA across the analyzed range, but there is in the microsatellites. I agree that there is in the microsatellites, and as stated in my comments I believe there is more than the authors present. I however, don’t agree that there is no difference over the range in the mtDNA: looking at the network, it is true that the Black Sea and TSS, not alone posses a haplotype/group. However it is clear that the frequency of haplogroups per area is not the same, and also that some areas don’t have specific haplotypes (e.g. is H4 and H5, with there surrounding haplotypes not present in TSS and the Black Sea). I think this is one point the authors need to consider, and of course if they disagree, I feel a stronger argumentation is warranted in the manuscript.

I have tried to add comments in an orderly fashion:

1) The title is very descriptive, but doesn’t sound all that good.

2) Paragraph from 64 to 72 doesn’t read very well.

3) Figure 1. It would be good to mark TSS on this figure. It is nicely done on Figure 2 (same thing could be done in figure 1).

I expect that the quality will be better with the final submission.

4) Line 111-113. I to some degree concur with the idea of using markers with higher mutation rate, however, suggesting to use microsatellites is less useful than going directly to more markers, e.g. whole genome shotgun sequencing.

5) Line 125. There is a space missing behind reference number 42. I think there is something missing at the end of that sentence. And the “n” before “populations” is probably a mistake.

6) Line 125-127. It is unclear to me how this is different from what you write in the sentence before? If you mean it can influence the genetic structure towards more differentiation, it would be good to spell out, as it is not clear.

7) Line 133. You don’t end the parentheses starting before “587”.

8) Line 134. When you refer to figure 1 here it would be helpful to include the names of these places on the map.

9) Line 135. There is something weird with the beginning of this line. If PLOS ONE allows I would write “XX et al (37) found”.

10) Line 136-137. For this point to hold, I think you should add in line 132-134 what genomic data was used. I presume it was CO1.

11) Line 140. Perhaps there is a better phrasing than “chaotic genetic patchiness”.

12) Line 154. I guess “A1”, is the first supplementary table, but this should be clearer. Also, it would be nice to do a count, so it’s obvious that there are 32 locations. There is also one point to many after A1.

13) Line 169. You may want to add one line about how they did fragment sizing at Cornell.

14) Line 174-177. You risk introducing a Wahlund effect by mixing potential populations, so an argument for this not happening would be merited somewhere. At the same time, I don’t see anything in the STRUCTURE plot supporting pooling samples in the way you are doing. I do however understand the necessity, I just don’t think that structure supports doing it.

15) Line 192-194. I am not a fan of choosing an optimal K in structure/admixture analysis, as all significant K’s are saying something about the data, and thus you are leaving out potential information. If the authors have not seen it, I would suggest the paper “A tutorial on how not to over-interpret STRUCTURE and ADMIXTURE bar plots”, written by one of the people who have done large amount of work on STRUCTURE (Daniel Falush). As a minimum I would argue that the authors should show that there are no signals in higher significant K´s.

16) Line 210-213. You don’t show the graph?

17) Line 237-238. Looks like you swapped the two last accession numbers?

18) Line 238. It is a little misleading to present it as 926 individuals, as you are effectively only adding 74 sequences – unless you are adding haplotypes multiple times so you are analyzing a total of 926 sequences? I would keep the part that with the 74 CO1 coming from 610, but write it came to a total of 390 individuals or individual sequences, or something along those lines.

19) Line 240. Think there is a point missing after “(70)”.

20) Line 241. You don’t close the parentheses before “e.g. (27)”.

21) Line 253. Sounds like you know that there will be signs of expansion. You could write “to detect potential population explansion”

22) Line 260. Do you mean the 20 populations that you pool the locations into, and what is the subpopulations here?

23) Line 268. Comma after “(S3 Table)” should be moved back one, and then a space.

24) Paragraph starting on line 271. Firstly, see my point 16. Secondly, I would also argue that there are clearly more signals in the PCoA than just two clusters. It is very clear that other samples are also clustering. I think marking more specifically the 20 “populations” is needed here. However, some indication of the 2 major clusters could be nice to keep.

25) Line 281. It is not clear what Table A1 is?

26) Line 282. Can I see from S2 Table that cluster C is fixed for Pm108? I thought S2 was showing proportion of null alleles?

27) Line 283-285. The S1 and S2 Tables I have show nothing about He. I don’t see the He values anywhere, these should be added.

28) Line 288. See my comments else where regarding the STRUCTURE inference. However, I still argue that you need to prove there is no significant meaningful higher structure.

29) Line 295. Again, see my comment about the PCoA elsewhere. But I will again argue you need to add information about all 20 “populations”.

30) Line 305 and other places. It is a bit confusing referring to the SoM and Dardanelles when they are not market on any map, nor in S1 Table.

31) Table 1. Could it be that those low values are due to population mixing (when you go from 32 sampling sites to 20 populations).

I realize that the values above the diagonal is ΦST , however you don’t state it.

32) Line 318. As the AMOVA test show more variation within groups than between groups I think this again indicates that more exploration of STRUCTURE and the PCoA is needed.

33) Table 2, 3, 5 and 6, please expand the legend to tell what SS is.

34) Line 337-338. As previous, is it actually 926 sequences you analyzed?

35) Line 356. Are the samples obtain from three different sources? It sounds like that, from your sampling, from Kalken, and then some extra Genebank sequences?

36) Figure 5. It would be good if you could add differences between haplotypes to the figure.

37) Line 361-362. It would be good to add to the table which populations are from the Mediterranean, Black Sea and so fourth.

38) Line 362. It is minor, but is that actually low compared to the length of the analyzed sequence?

39) Line 371. Again, it would be beneficial to add to table 4 which subpopulations are from TSS.

40) Table 4. It doesn’t seem consistent to me what the pop code, population and subpopulation is referring to?

41) Line 385-389. It seems that mtDNA is supporting better 3 clusters than just 2.

I’m fine with you comparing them as 20 groups, but from the STRUCTURE plot they are obviously not genetic populations. So you should be careful with the term populations.

42) Line 396. Missing space after the parentheses.

43) Line 398. I don´t see S1 table showing anything about haplotypes?

44) Line 432. It is very hard to follow the location names, when some are not added to the map.

45) Line 440-442. I’m not sure what you are trying to say with this sentence, so if you could clear up what the point is, that would be helpful.

46) Line 444-445. This is very speculative, and not possible to say with this data. I think it’s fine to keep, as it is interesting, but it should at least be stated that you have no possibility of investigating that with this data.

47) Line 450. I’m still not convinced it is just 2 populations, and thus as suggested earlier, this should be analyzed further.

48) Line 453. What recent geneflow wouldn’t affect the microsatellites, but only the mtDNA?

49) Line 475. You need to shortly state what reference 47 was analyzing (species/markers), otherwise it’s hard to understand.

50) Line 521-525. “Been replaced” sounds weird. Rather something along the lines of “have become dominant, due to drift or selection”, or something like it.

I don’t know what form the journal want, but you should look over the references as they are not consistent in how you write. E.g. “found by (47)” vs “Paterno et al. (10)”.

Location 19 has a much higher proportion of green population than the locations closer to the TSS. How do you explain this if green is just flowing from the black sea, and should thus arguably become less and less dominant the further you go into the Mediterranean?

Figure 5. If you can make the labels a bit bigger, it would be helpful to read the location names. The figure would benefit with differences between haplotypes/groups.

Supplementary figure 1, I think is missing a legend.

Reviewer #2: This study examined the population genetic structure of the marbled rock crab Pachygrapsus marmoratus, focusing on the Turkish Strait System between the Black Sea and the Mediterranean Sea. I think the topic in this study is interesting and important in the population genetics of coastal organisms. However, the manuscript is not well prepared, and thus extremely hard to read and/or understand tables, figures, and arguments proposed by the authors. This is because there are many flaws in the presentation of results, and datasets and analyses are insufficient to support the arguments. I think the English writing should be improved too. I left specific comments on the attached PDF directly.

Reviewer #3: The study examined population genetic structure of the marbled rock crab Pachygrapsus marmoratus in the TSS, a transition zone between the Black Sea and Mediterranean Sea using mtDNA and microsatellite markers.  Microsatellite data from 5 loci showed marked genetic differentiation with 2 genetic clusters exhibiting differential distribution between the Black Sea and the Mediterranean.  Mitochondrial CoI sequences reveal the occurrence of 3 common haplotypes, which while exhibiting low sequence divergence values exhibit geographically different distributions broadly concordant with a genetic break between the Black Sea and the Mediterranean.

While the topic is of broad interest to the fields of phylogeography and population genetics with particular attention to admixture zones, I have several comments/clarifications that need to be addressed before the paper is acceptable for publication.

Major comments:

- Sample information. Suggest to include the information on Basin (Table S1) in Figure 1 perhaps as part of the Figure caption, for easier reference.

- l.247 Indicate reason for excluding individuals with STRUCTURE q<0.7 from pairwhise Phi-ST analysis?

- l.304. Indicate overall Fst value and associated p-value (microsatellites) to support statement of significant differentiation among sampling sites

- L.305. On the Dardanelles samples - indicate population number or code at the first instance for easy reference.  The detail for the Dardanelle samples being population #10 was only mentioned later in line 370.

- Table 1. For correction, the table caption indicates "nuclear genetic divergence", but it includes Phi-ST values from mitochondrial data

- Table 1. Suggestions: (1) Indicate Basin or Group (Black Sea, TSS, Mediterannean etc...) on Table 1 for easier reference and (2) Might be more visually informative to make this into a heatmap (retaining the values for reference).

- AMOVA. Table 2 and 3 can be combined in a single table.

- l.350. H7 is not geographically restricted to the Black Sea, as it is found in the Mediterranean samples

- On genetic structure, it is important to state first whether the mitochondrial data reveal significant genetic structure (Phi ST > 0), before presenting the AMOVA results. Also, what is the overall Phi-ST value and its associated p-value?

- l.385. The same two hypothesis of geographic structure tested with microsatellites

- AMOVA Tables 5 and 6 can be combined

On comparison between microsatellite and mt CoI results:

- On line 453, the authors report that microsatellites and mtDNA did not reflect the same degree of divergence in the Mediterranean. What does 'degree of divergence' mean? Are they referring to the Fst and Phi-st values? It will be difficult to compare divergence for different marker types. If anything, what is clear is that there is a broadly concordant pattern of geographic structure revealed by both markers. Marked genetic groups for the microsatellite data is clear (clusters M and C, with differential distributions across the sampled range). For the mtDNA data, yes the sequence divergence may be low (with closely related haplotypes separated by 1-2 bp only), however, their markedly different geographical distributions (H7 predominant in the Black Sea and TSS, H2 and H4 predominent in the Mediterranean and Atlantic) also give rise to a geographic break broadly consistent with the microsatellite data. This is also seen in the AMOVA for both markers (Tables 3, 4, 5 and 6).

- I suggest that the authors present the cytonuclear pattern (microsatellite cluster - mtDNA haplotype combination) to gain further insight into the patterns of admixture in the TSS. To what extent do individuals with belonging to different microsatellite clusters share mitochondrial haplotypes?

- Suggest to perform isolation with migration (IMA) analyses to test hypotheses regarding mitochondrial gene flow between microsatellite clusters. it might be possible to test hypotheses regarding directionality of mitochondrial gene flow, as well as infer whether there was gene flow after divergence (or none).

Minor comments/corrections

- l.70 North Sea-Baltic Sea

- l.125 connectivity among n populations

- l. 154. ... in A1. Do you mean S1 Table?

- l.265 "didn't" replace with "did not"

6. PLOS authors have the option to publish the peer review history of their article (what does this mean?). If published, this will include your full peer review and any attached files.

Reviewer #1: **Yes: **Charles Christian Riis Hansen

Reviewer #2: No

Reviewer #3: No

---

## [Author Response · Author response to Decision Letter 0]

9 Jan 2022

Responses to general editorial comments:

1. The frequency of mt haplogroups differs by areas.

This message is now emphasized more throughout the manuscript.

2. I would argue that the authors should show that there are no signals in higher significant K´s: 

Now the plots for K=3, K=4 and K=5 are added as supplementary figure S2 Fig, There is not a pattern interpretable for K values other than 2. At K=3, some individuals belonging to Cluster C form the third cluster but this doesn’t correspond to any geographical pattern. 

3. Present the cytonuclear pattern (microsatellite cluster - mtDNA haplotype combination) to gain further insight into the patterns of admixture in the TSS. 

We added a Supplementary Figure (S4 Fig) showing both main haplogroups of mtDNA and clusters of microsatellites. Here, it is seen that in the Black Sea and the TSS, individuals from Cluster C share the same mitochondrial haplogroup (H7, which is the dominant haplotype in the Black Sea) whereas in the sympatric clusters in the Mediterranean, haplogroups do not always correspond to microsat clusters.

4. Perform isolation with migration (IMA) analyses to test mitochondrial gene flow between clusters. 

We run IMA several times under different conditions. However, the results we were obtaining, although similar, were in the range of million years for the splitting time. As such they could not be interpreted. Probably using only one, and relatively short, marker could not provide enough information for the meaningful results.

5. Get a native English speaker to help with language, either an academic or certified English copy-editor.

6. Supplementary figures and tables, add titles and more detailed legends and descriptions.

We added more figures/tables to supplementary and improved descriptions of figures&tables.

Reviewer #1: Comments PLOS ONE

The authors analyze microsatellite and mtDNA data from the marbled crab in the Turkeys, Black and Mediterranean Sea. The analysis are restricted to 5 microsatellites and the CO1 region of the mitochondrial DNA. The authors find a genetic difference between the Black and Turkeys Sea on one side and the Mediterranean Sea on the other when looking at the microsatellite data, but not when analyzing the mtDNA data.

Overall, I find this study quite interesting, and a presumable need for further knowledge about the crab. It would have been nice to see more independent markers, but I also realize you sometimes work with what you have. But it would be nice to see some SNP data for these crabs in the future.

I am not sure to what extend I agree about the overall conclusion of the paper. The author states there is no genetic divergence in the mtDNA across the analyzed range, but there is in the microsatellites. I agree that there is in the microsatellites, and as stated in my comments I believe there is more than the authors present. I however, don’t agree that there is no difference over the range in the mtDNA: looking at the network, it is true that the Black Sea and TSS, not alone posses a haplotype/group. However it is clear that the frequency of haplogroups per area is not the same, and also that some areas don’t have specific haplotypes (e.g. is H4 and H5, with there surrounding haplotypes not present in TSS and the Black Sea). I think this is one point the authors need to consider, and of course if they disagree, I feel a stronger argumentation is warranted in the manuscript.

General Response to Reviewer 1:

Thank you very much for very detailed and very helpful comments and corrections to the manuscript. 

We also agree that there is a strong geographical pattern in distributions of haplotypes in mtDNA. Specifically, H7 is the dominant haplotype in the Black Sea and its surrounding haplotypes are only found in the Black Sea or in the TSS. On the other hand, as is suggested here, surrounding haplotypes of H4 and H5 are not present in the Black Sea. We now made this point more clear in the manuscript.

I have tried to add comments in an orderly fashion:

1) The title is very descriptive, but doesn’t sound all that good.

Title is changed now. 

2) Paragraph from 64 to 72 doesn’t read very well.

Thank you, now we made changes on that part and merged it with the next paragraph. The new paragraph now reads as follows: 

“These transition zones are often regions where divergent lineages meet again in secondary contact. Such secondary contact has been observed in species with very different life histories and taxonomic groups. Some examples include blue mussel (Mytilus edulis/trossulus) (12), Balthic clam (Limecola balthica) (8), vase tunicate (Ciona intestinalis) (13), European flounder (Platichthys flesus) (14), European sea bass (Dicentrarchus labrax) (11) among many others (reviewed in (15)). Despite the presence of several studies investigating the role of other transition zones in the Mediterranean in gene flow among locations [5,11,12], studies on the TSS are limited. The TSS (Fig 1) is a transition zone between the Black Sea and the Mediterranean Sea by means of two relatively narrow straits (the Bosphorus Strait and the Dardanelles) and an almost entirely landlocked sea (the Sea of Marmara, SoM). The Black Sea has been repeatedly connected and separated from the Mediterranean Sea through the opening and closure of the Dardanelles and Bosphorus Strait during the Quaternary period climatic fluctuations [13–15]. These fluctuations have a role in shaping the genetic diversity and diversification in and around the system [16–18]. In the Black Sea, for example, populations of various taxa have lower genetic diversity in comparison to those in the Eastern Mediterranean, due to founder effects and/or genetic drift in line with their more recent establishment [19–21]. The most recent connection between the Mediterranean and the Black Sea was established around 8400 BP, with an inflow of its waters to the Black Sea, and TSS took its current form in terms of its hydrology and current systems around 7500 BP [13,14]. The current hydrology of the TSS is characterized by strong stratification (halocline and thermocline) and a two-layer current system, in which brackish water from the Black Sea flows to the Aegean above the denser saline Mediterranean waters that flow towards the Black Sea [22–24]. “

3) Figure 1. It would be good to mark TSS on this figure. It is nicely done on Figure 2 (same thing could be done in figure 1). 

Now TSS is also marked in Figure 1.

I expect that the quality will be better with the final submission.

4) Line 111-113. I to some degree concur with the idea of using markers with higher mutation rate, however, suggesting to use microsatellites is less useful than going directly to more markers, e.g. whole genome shotgun sequencing.

We also agree that more markers would be more useful in making inferences about phylogeography and/genetic structuring and give more clear signals. On the other hand, several studies found similar overall patterns in terms of genetic structure and genetic diversity by using either SNP data or microsatellites (see Sunde et al. 2020). 

5) Line 125. There is a space missing behind reference number 42. I think there is something missing at the end of that sentence. And the “n” before “populations” is probably a mistake.

Corrected accordingly.

6) Line 125-127. It is unclear to me how this is different from what you write in the sentence before? If you mean it can influence the genetic structure towards more differentiation, it would be good to spell out, as it is not clear.

Now it is clearly stated that fronts can lead to more differentiation. The text now reads as follows: “On the other hand, due to its long planktonic larval duration, the influence of oceanographic fronts can lead to strongly influence its genetic structure into differentiation in the genetic structure of the marbled crab [17].”

7) Line 133. You don’t end the parentheses starting before “587”.

Corrected accordingly.

8) Line 134. When you refer to figure 1 here it would be helpful to include the names of these places on the map.

Figure 1 already includes labels for Black Sea and the Mediterranean but most of the Atlantic Ocean and Canary Islands are not covered by the map in this version. Therefore, we removed the reference to Figure 1.

9) Line 135. There is something weird with the beginning of this line. If PLOS ONE allows I would write “XX et al (37) found”.

Changed accordingly.

10) Line 136-137. For this point to hold, I think you should add in line 132-134 what genomic data was used. I presume it was CO1.

Changed accordingly.

11) Line 140. Perhaps there is a better phrasing than “chaotic genetic patchiness”.

We used this term as it was mentioned and discussed several times in papers about this species (Iannucci et al. 2020; Silva et al. 2009).

12) Line 154. I guess “A1”, is the first supplementary table, but this should be clearer. Also, it would be nice to do a count, so it’s obvious that there are 32 locations. There is also one point to many after A1.

Both corrections are made accordingly.

13) Line 169. You may want to add one line about how they did fragment sizing at Cornell.

We now added more information here about fragment sizing: “Fragment sizes were scored on an Applied Biosystems 3730xl DNA Analyzer”

14) Line 174-177. You risk introducing a Wahlund effect by mixing potential populations, so an argument for this not happening would be merited somewhere. At the same time, I don’t see anything in the STRUCTURE plot supporting pooling samples in the way you are doing. I do however understand the necessity, I just don’t think that structure supports doing it.

We replaced those analyses by new ones, excluding sites with sample sizes less than 4. We decided to do that to avoid the risk of Wahlund effect and also due to the difficulty of justifying the reasoning. 

15) Line 192-194. I am not a fan of choosing an optimal K in structure/admixture analysis, as all significant K’s are saying something about the data, and thus you are leaving out potential information. If the authors have not seen it, I would suggest the paper “A tutorial on how not to over-interpret STRUCTURE and ADMIXTURE bar plots”, written by one of the people who have done large amount of work on STRUCTURE (Daniel Falush). As a minimum I would argue that the authors should show that there are no signals in higher significant K´s. 

Thank you, we have inspected patterns in other K values between 2 and 5. Since there was not any pattern that is interpretable for other K values, we went on with K=2 interpretations. Barplots showing K=2, K=3, K=4, and K=5 are added in the supplementary figure. 

16) Line 210-213. You don’t show the graph?

Now the corresponding graph is referenced (S1 Fig) here. 

17) Line 237-238. Looks like you swapped the two last accession numbers?

Changed accordingly.

18) Line 238. It is a little misleading to present it as 926 individuals, as you are effectively only adding 74 sequences – unless you are adding haplotypes multiple times so you are analyzing a total of 926 sequences? I would keep the part that with the 74 CO1 coming from 610, but write it came to a total of 390 individuals or individual sequences, or something along those lines.

For constructing the network, we analysed 74 COI haplotypes from GenBank but in order to reflect sample sizes of previous studies, we adjusted the width of pies accordingly. Thus, in the network analysis, information corresponding to a total of 926 individuals was used. So, writing that a total of 390 individuals would be misleading for the network analysis part in the manuscript.

19) Line 240. Think there is a point missing after “(70)”.

Point added accordingly.

20) Line 241. You don’t close the parentheses before “e.g. (27)”.

Changed accordingly.

21) Line 253. Sounds like you know that there will be signs of expansion. You could write “to detect potential population explansion”

Changed accordingly.

22) Line 260. Do you mean the 20 populations that you pool the locations into, and what is the subpopulations here?

Here, we meant 20 sampling sites (also referred as populations through the manuscript). Since we are not pooling populations any more in this version of the manuscript, we hope it became easier to follow.

23) Line 268. Comma after “(S3 Table)” should be moved back one, and then a space.

Changed accordingly.

24) Paragraph starting on line 271. Firstly, see my point 16. Secondly, I would also argue that there are clearly more signals in the PCoA than just two clusters. It is very clear that other samples are also clustering. I think marking more specifically the 20 “populations” is needed here. However, some indication of the 2 major clusters could be nice to keep.

There was no signal for grouping of other sampling sites but now we added a supplementary figure to show how “sampling sites” group in this pcoa plot (S5 Fig).

25) Line 281. It is not clear what Table A1 is?

Thank you, it was meant to refer to Figure 4 where we show allelic distributions of microsatellite loci. Now it is corrected.

26) Line 282. Can I see from S2 Table that cluster C is fixed for Pm108? I thought S2 was showing proportion of null alleles?

Here, we also referred to the wrong table/figure. It refers to Figure 4 and S4 Table and now it is corrected.

27) Line 283-285. The S1 and S2 Tables I have show nothing about He. I don’t see the He values anywhere, these should be added.

Thank you very much. Yes, now we added a separate table with He values as supplementary Table 4 and the correct reference is added here.

28) Line 288. See my comments elsewhere regarding the STRUCTURE inference. However, I still argue that you need to prove there is no significant meaningful higher structure. 

DeltaK method suggested K=2 but probability by K method suggested highest probability at K=5 based on median values of Ln(Pr Data). (S3 Fig). However, STRUCTURE graphs for other K values (K: 3-5) did not give any meaningful structure (S2 Fig). 

29) Line 295. Again, see my comment about the PCoA elsewhere. But I will again argue you need to add information about all 20 “populations”.

See our answer above (comment #24).

30) Line 305 and other places. It is a bit confusing referring to the SoM and Dardanelles when they are not market on any map, nor in S1 Table.

Now the SoM, Dardanelles, and the Bosphorus Strait are marked in Figure 1.

31) Table 1. Could it be that those low values are due to population mixing (when you go from 32 sampling sites to 20 populations).

We don’t believe this to be the case. Since we are now not pooling any populations and the pattern is still the same, it doesn’t seem to be an artefact.

I realize that the values above the diagonal is ΦST , however you don’t state it.

Thank you, now we stated it in the title of the Table 1 (now it is Figure 3).

32) Line 318. As the AMOVA test show more variation within groups than between groups I think this again indicates that more exploration of STRUCTURE and the PCoA is needed. 

As explained in our answers to the comments #24 and #28, there was not any interpretable pattern with more detailed look at both STRUCTURE and PCoA. This is also consistent with previous studies of the marbled crab using microsatellites where they found the largest variation within groups (see Ianucci et al. 2020, Silva et al. 2009).

33) Table 2, 3, 5 and 6, please expand the legend to tell what SS is.

Explanation is now added to corresponding tables.

34) Line 337-338. As previous, is it actually 926 sequences you analyzed?

Yes, just for the Network analyses, we used 926 sequences (manually added) for better representation of sample sizes.

35) Line 356. Are the samples obtain from three different sources? It sounds like that, from your sampling, from Kalken, and then some extra Genebank sequences?

Yes, some of the individuals used in this study were already sequenced in Kalkan et al. 2013. Extra Genbank sequences were added for network analysis.

36) Figure 5. It would be good if you could add differences between haplotypes to the figure.

Since each line corresponds to one mutation difference as explained in the figure legend, we do not see any reason to include this information. We now mention the divergence in percentage between main haplotypes in the text: “The network showed a star-like pattern with only two bp separating the three most common haplotypes (H2, H4 and H7), corresponding to 0.4 % divergence. “

37) Line 361-362. It would be good to add to the table which populations are from the Mediterranean, Black Sea and so fourth.

An additional column of basins is now added to Table 4.

38) Line 362. It is minor, but is that actually low compared to the length of the analyzed sequence?

Thank you, here we were referring to mostly low nucleotide diversity so this comment (“low) is now removed from this sentence. The new sentence now reads as follows: “Among the 31 haplotypes included in the dataset sampled in this study, there were 64 polymorphic sites with a total of 67 mutations.”

39) Line 371. Again, it would be beneficial to add to table 4 which subpopulations are from TSS.

An additional column of basins/regions are now added to Table 4.

40) Table 4. It doesn’t seem consistent to me what the pop code, population and subpopulation is referring to?

Now “pop code” is changed to “code” and “subpopulation” is changed to “sampling site”. “Population” in the title is changed to “clusters”.

41) Line 385-389. It seems that mtDNA is supporting better 3 clusters than just 2.

Yes, it is true that mtDNA supports 3 geographical regions better than 2 clusters. The order is changed in the sentence accordingly.

I’m fine with you comparing them as 20 groups, but from the STRUCTURE plot they are obviously not genetic populations. So you should be careful with the term populations.

Thank you. We tried to avoid using the term “populations” for sampling sites in general. Now they are changed to sampling sites.

42) Line 396. Missing space after the parentheses.

Corrected accordingly.

43) Line 398. I don´t see S1 table showing anything about haplotypes?

We now added a supplementary Figure (Fig 6) which shows haplotype distributions of different sampling sites.

44) Line 432. It is very hard to follow the location names, when some are not added to the map.

All of the mentioned names are now added to the map.

45) Line 440-442. I’m not sure what you are trying to say with this sentence, so if you could clear up what the point is, that would be helpful.

This part in Discussion where we discuss Dardanelles samples in detail is now removed as it is too speculative. The sentence referred in this comment (Line 440-442) is therefore also removed now.

46) Line 444-445. This is very speculative, and not possible to say with this data. I think it’s fine to keep, as it is interesting, but it should at least be stated that you have no possibility of investigating that with this data.

This part in Discussion where we discuss Dardanelles samples in detail is now removed as it is too speculative. The sentence referred in this comment (Line 444-445) is therefore also removed now.

47) Line 450. I’m still not convinced it is just 2 populations, and thus as suggested earlier, this should be analyzed further.

See answers to comments #24, #28, and #32.

48) Line 453. What recent geneflow wouldn’t affect the microsatellites, but only the mtDNA?

Recent gene flow was removed as an explanation for similarity in mtDNA in the Mediterranean as incomplete lineage sorting of ancestral polymorphism seems a more plausible explanation.

49) Line 475. You need to shortly state what reference 47 was analyzing (species/markers), otherwise it’s hard to understand.

Since this scenario where cluster C was the local population is now removed, this sentence in this part is also not included.

50) Line 521-525. “Been replaced” sounds weird. Rather something along the lines of “have become dominant, due to drift or selection”, or something like it.

We decided to remove that sentence it was too speculative.

I don’t know what form the journal want, but you should look over the references as they are not consistent in how you write. E.g. “found by (47)” vs “Paterno et al. (10)”.

References are edited accordingly.

Location 19 has a much higher proportion of green population than the locations closer to the TSS. How do you explain this if green is just flowing from the black sea, and should thus arguably become less and less dominant the further you go into the Mediterranean?

We also noticed this pattern and we don’t have a definite explanation for this pattern. It could be a possibility that larvae of the marbled crab have been transported by ballast water of ships going from the Black Sea&TSS into Levantine coasts.

Figure 5. If you can make the labels a bit bigger, it would be helpful to read the location names. The figure would benefit with differences between haplotypes/groups.

Label sizes were adjusted accordingly for easier reading of different geographical regions. The difference between haplotypes (groups) is only 1 mutation which was also mentioned in the figure caption. The percentage of difference between main haplogroups is now added to the Results (but not directly to the Figure as explained in our answer to comment #36).

Supplementary figure 1, I think is missing a legend.

Legend of Supplementary Figure 1 and other recently added supplementary Figures/Tables are now included.

Reviewer #2: This study examined the population genetic structure of the marbled rock crab Pachygrapsus marmoratus, focusing on the Turkish Strait System between the Black Sea and the Mediterranean Sea. I think the topic in this study is interesting and important in the population genetics of coastal organisms. However, the manuscript is not well prepared, and thus extremely hard to read and/or understand tables, figures, and arguments proposed by the authors. This is because there are many flaws in the presentation of results, and datasets and analyses are insufficient to support the arguments. I think the English writing should be improved too. I left specific comments on the attached PDF directly.

1) Line 4. I think COI shows genetic divergence between populations of the Black Sea and Mediterranean too.

Yes, COI does show the divergence between Black Sea and the Mediterranean but the divergence in sympatric clusters in the Mediterranean was only captured by microsatellites. However, the title is changed now to “Mitonuclear genetic patterns of divergence in the marbled crab, Pachygrapsus marmoratus (Fabricius, 1787) along the Turkish seas”.

2) Line 5. The use of "sympatric populations" is not appropriate here. Please refer to my comments on Line 461.

This sentence is now changed and we “sympatric populations” term is not used.

3) Line 39. clusters

Changed accordingly.

4) Line 40. individuals belonging to cluster

Changed accordingly.

5) Line 41. please rephrase this

This part is rephrased as follows: 

“While individuals from the cluster C are present in all the sampling locations, those from the cluster M are only detected along the Mediterranean coasts including the Aegean and the Levantine basins. “

6) Line 47. each other at this stage.

Changed accordingly.

7) Line 53. "Transition zones" of what?

Now it is changed to “Transition zones between different biogeographical regions”. 

8) Line 57-62. These examples are hard to follow for readers not familiar with the Mediterranean if there are no figures indicating areas.

Now, we updated Figure 1. It includes all of the mentioned areas for an easier understanding and we added the reference to this paragraph.

9) Line 70. Please don't use italic for parentheses throughout the manuscript.

Changed accordingly.

10) Line 76. The abbreviation of this was shown above.

Changed accordingly.

11) Line 81-82. Please add references for this sentence.

References added accordingly.

12) Line 85. connection between the Mediterranean and the Black Sea

Changed accordingly.

13) LIne 85. BP?

Thank you, corrected accordingly.

14) Line 97. I think the two-layered current does not decrease gene flow but increases one-directional transportation of propagules.

We agree that it increases one-directional transport of larvae (from Black Sea to Aegean) but it can also limit the transport in the opposite direction as explained in the sentences below, as follows: “While the direction of the upper layer current increases one-directional transport of larvae (from the Black Sea to the Aegean), it can also limit the transport in the opposite direction as found out in studies of some marine species, such as anchovy…”

15) Line 103. and the marbled crab,

Added accordingly.

16) Line 104. XXX et al. and XXX et al. (35, 38)

Changed accordingly.

17) Line 105. two

Changed accordingly.

18) Line 106. I did not understand why and how the spatial distribution of type II and III were influenced by the two-layered current regime from this sentence. Please clarify.

Thank you, now we added “ the absence of type II in the Black Sea” to this sentence, as the two-layered current regime might have contributed to its absence in the TSS and the Black Sea.

19) Line 107. Is this a temporary exchange? or continuous one? Please clarify if possible.

Here, we mean the establishment of the two-layered current pattern in the TSS. This exchange has been continuous since around 7500 years. Now, it is added to the sentence, and the sentence reads as follows: “… recently established water exchange through the TSS which took its current form around 7500 BP (see [22,24]) might have created secondary contact zones between previously separated lineages in the Black Sea and the Mediterranean. “

20) Line 107. through?

Changed accordingly.

21) Line 116. Please add information on the range of the geographical distribution of this species.

Its distribution range is now added to this paragraph.

22) Line 122. Please add the range of salinity.

The species was found to survive starting from salinity level of 15 ‰ to 35‰ in an experimental study (Karadal 2018). This range was also added to the text. 

23) Line 123. Please add the range of salinity.

Corresponding salinity ranges of both the TSS and the Mediterranean are now added to this part.

24) Line 124-125. I think this sentence is incomplete.

Thank you, the sentence was indeed not complete and now it is corrected.

25) Line 130. Please indicate the name of the straits in Fig. 1. 

Name of the straits and the SoM are now added to Fig. 1

26) Line 135. XX et al. (37)

This reference was changed accordingly.

27) Line 154. Did you mean Table S1?

Yes, we meant Table S1, it is corrected accordingly.

28) Which Taq? Please clarify.

The details of Taq are now added (Thermo Scientific).

29) Line 174-176. I did not agree with this treatment because this may fail to detect distinct populations that are not differentiated by microsatellites. I recommend you exclude populations with a limited number of specimens.

Populations with less than four individuals are now excluded and analyses were repeated accordingly.

30) Line 176. Table S1? Pleas add number of specimens of each population in Table S1.

A new column with sample sizes is now added to Table S1.

31) Line 185. Please include this in Supplementary data.

This was included in Supplementary Table 3 and now we added its reference to this sentence as well.

32) Line 191. Are these the numbers of MCMC repeats? Please clarify burn-in lengths.

Burn-in lengths were also the same with MCMC numbers, and this information is now added in parentheses.

33) Line 193. Something wrong with this sentence. Please rephrase this.

Thank you, rephrased sentence now reads: “Best delta K value was estimated using the method of Evanno et al. (2005) as implemented in STRUCTURE HARVESTER.”

34) Line 245. I think you should show pairwise FST of COI because the genetic difference between the Mediterranean and the Black Sea may be revealed, considering the results of haplotype network.

Pairwise Fst values of mtDNA COI was included in the lower horizontal part of Table 1 (now Fig 3). It indeed shows the differentiation between Mediterranean and the Black Sea/TSS as the main pattern.

35) Please clarify why you did this treatment.

Now we did not exclude any individuals and repeated the Fst analyses. 

36) Line 257. Pleas add a Table with Ne. He, Ho, and Ar of each sampling locality.

Thank you very much for pointing this out. Although we refer to these values in the text, the table itself was missing. We now added the table with these diversity metrics. It is in the supplementary Table S4.

37) Line 271. Did you mean K = 2 was supported by the analysis of HARVESTER? Please clarify.

Yes, that was what we meant with this sentence. Now the sentence was reframed as “The most likely number of clusters was evaluated with the ΔK method and results supported two clusters.”

38) Line 272-273. Is this true? I see the red lines in the individuals of the Black Sea in Fig. 2, although it is hard to recognize due to the low resolution of the figure.

Yes, this is true. There are no lines in the Black Sea but some individuals have up to 5 % of cluster M assignment . We tried to increase the resolution with the new figure.

39) Line 281. Is this reference correct?

Thank you, it was not correct. The correct reference was Figure 5 and now it is corrected.

40) Line 283. It is extremely hard to understand this sentence and Table S2. Please add locality and cluster names in the Table S2.

Table S2 which shows the null alleles, was the wrong reference, we now cite Figure 5 and remove the details about sampling sites.

41) Line 284. Is this correct?

Thank you, it was not correct. The correct reference was again Figure 4 and now it is corrected.

42) Line 285. ??? 

Here, the reference was corrected to S4 Table which now includes diversity values of structures clusters and sampling sites.

43) Line 305-308. First of all, Dardanelles (n = 7) is too small to discuss population genetic structure. In addition, Table 1 (FST) does not sufficiently support that the Dardanelles (population 10) is differentiated from those of the Mediterranean (population 11-20). 

We now added the warning that sample size is too small for detecting patterns (N=7 and N=10 for mtDNA and microsatellites, respectively. It is clear in our heatmap (now called Fig 3) that Dardanelles is not differentiated from the Mediterranean but rather from the Black Sea. We hope that this part is now clearer.

44) Line 309-311. This result supports that you should not merge populations (Line 174-176) with small sample size.

Thank you, yes, our updated analyses does not include merged populations.

45) Line 318. I recommend you to do SAMOVA analyses to detect subgroups used for AMOVA.

We had previously run Geneland software which also includes geographical information and we had found two groups separated by the Dardanelles (Black Sea&TSS and the Mediterranean). Since we already test the effect of geographical regions and also STRUCTURE clusters on genetic structure, we think SAMOVA won’t add anything new to our interpretation.

46) Line 347. Did this study cover the entire range of known geographic distribution of P. marmoratus?

The mentioned network analysis covers most of its distribution range. Now the sentence was reframed accordingly.

47) Line 348-350. I did not understand the purpose of this sentence.

This information about geographical distribution of the main haplotypes was requested by a previous reviewer. Those parts can be removed if it is really necessary but we decided to keep them for clarification of the patterns in the network.

48) Line 355. I did not understand the border between East and West Mediterranean.

This border is defined according to oceanographic properties of the region (see for example "Physical Oceanography of the Eastern Mediterranean Sea, Akpinar et al. 2016 and the references therein).

49) Line 356. GenBank

Thank you, corrected accordingly.

50) Line 369-371. I think the sample number (n = 7) is too small to discuss haplotype and nucleotide diversity. And, it is hard to tell which station belongs TSS from Table 4. 

Table 4 now includes the information about basins. Since we also agree that our sample size is not large enough, we removed that section.

51) Line 375. Table 1. Please add a column of regions (e.g., Aegean Sea, Black Sea, TSS, and Mediterranean). 

Regions are now shown as different colors for corresponding regions of the sampling sites in Table 1 (now Fig3).

Line 385-386. I think pairwise FST analysis of COI also helps to understand population genetic structure of P. marmoratus. Please add a table like Table 1.

Pairwise FST analysis of COI was included in the below horizontal part of Table 1 (now Fig 3).

52) Line 401-403. Did you mean the genetic structure of the Dardanelles population was more similar to the Mediterranean populations than TSS? If so, Table 1 is the results of microsatellites and any other results on mtDNA shown in the manuscript does not support this argument.

As mentioned above, pairwise FST values of mtDNA COI was included in the below horizontal part of Table 1 (now Fig 3). It shows no differentiation between Mediterranean and the Dardanelles samples (although sample size of Dardanelles is small to make strong inferences).

53) Line 432-434. This is not obvious from the results of the current study due to the small sample size of Dardanelles and any results on mtDNA did not support the idea in this sentence. The haplotype network shows that mainly the same haplotype (H7) is dominant in the TSS (Dardanelles was not distinguished) and the Black Sea. 

As also explained above, below horizontal part of the Table 1 (now Fig 3) supports this idea. We also agree that its sample size is not enough for a strong inference and a warning is added here. 

54) Line 444-445. This needs more detailed information on both the salinity in these areas and the tolerance of P. marmoratus. Please add the salinity of Aegean and several localities in TSS at least.

Since this section about Dardanelles was removed from Discussion, this sentence does not exist anymore in the revised manuscript. Howevert more information about salinity levels was added as follows: Whereas the salinity of Black Sea Water (coming from Black Sea to Mediterranean in the upper layer) is around 16-18 psu, the Mediterranean water entering from the Dardanelles to the Sea of Marmara through the lower current has a salinity of around 38-39 psu (Ünlüata et al. 1990; Gündüz and Özsoy 2016). Salinity tolerance of P. marmoratus was added to Introduction.

55) Line 451-453. There are clear differences present between the population in the Mediterranean and the Black Sea. Only one haplotype was dominant in the Black Sea. 

We agree that there is a clear difference between these regions. However, here, we were referring to the pattern in the Mediterranean. Now, we made this part more detailed for a clearer understanding:

“In contrast to divergence in microsatellites, mtDNA data did not reflect the same degree of divergence in the sympatric clusters in the Mediterranean. Visual representation of mtDNA haplogroups combined with microsatellite clustering of individuals (S4 Fig) reveal that in the Black Sea and the TSS, individuals from cluster C share the same mitochondrial haplogroup (H7, which is the dominant haplotype in the Black Sea) whereas in the sympatric clusters both clusters share similar haplogroups in the Mediterranean . “

56) Line 453-454. More detailed explanation is needed here. Recent gene flow affects the population genetic structure of both nuclear and mitochondrial genes.

Here, recent gene flow as an explanation for similarity in mtDNA was removed.

57) Line 453. The inconsistency between nuclear and mitochondrial genes also results from sex-biased migration (but this may be not common in marine invertebrates) and the difference in the effects of genetic drift due to the smaller effective population size in mitochondrial genes. 

Sex-biased migration probably doesn’t apply to crabs. On the other hand, smaller effective population of mtDNA would result more differentiation in mtDNA rather than microsatellites, as opposed to what we observe in the Mediterranean. Maybe a comment for inconsistency.

58) Line 461. "Common" or "Mediterranean" are not populations but clusters suggested by STRUCTURE analysis.

Thank you, now they were changed to “clusters” instead of populations.

59) Line 472-473. This does not support the idea. You have to compare He between the Black Sea and Mediterranean populations, not two clusters suggested by STRUCTURE since "C" in the Mediterranean did not experience founder effects due to invasion into the Black Sea.

This is now corrected.

60) Line 473-474. This too.

Changed/corrected accordingly.

61) Line 473-475. This sentence is nonsense. Fratini et al. used longer COI sequences thus higher Hd in their analysis is natural. In addition, I did not understand the reason why you compared the Hd of the two studies.

This part is now removed.

62) Line 482-483. Do you think that gene flow between sympatric cluster C and M are basically restricted? Is this idea supported by which results?

This part is now removed.

63) Line 484-485. Which data implies the range expansion in the Mediterranean after the invasion of M. marmatus into the Black Sea?

This part is also now removed from the Discussion as we decided this scenario involved too many assumptions that we could not support with the current data. 

64) Line 507. Please clarify how you calculated this.

 Here, 15.000 years refer to the duration of Last Glacial Maximum (starting around 22.000 ya and ending around 7.000 ya).

65) Line 521-522. How?

Now we removed that part (it seemed too hypothetical).

66) Line 527-528. I think your opinions on mtDNA are inconsistent thorough the manuscript (e.g., this sentence vs. Line 451-453).

We now tried to make our arguments on mtDNA clearer. First, we agree that there is a strong geographical signal in CO1. Secondly, as can also be seen in supplementary figure 4, both nuclear clusters in the Mediterranean share a similar pattern of haplotypes and we argue that this is due to incomplete lineage sorting of CO1.

Reviewer #3: The study examined population genetic structure of the marbled rock crab Pachygrapsus marmoratus in the TSS, a transition zone between the Black Sea and Mediterranean Sea using mtDNA and microsatellite markers. Microsatellite data from 5 loci showed marked genetic differentiation with 2 genetic clusters exhibiting differential distribution between the Black Sea and the Mediterranean. Mitochondrial CoI sequences reveal the occurrence of 3 common haplotypes, which while exhibiting low sequence divergence values exhibit geographically different distributions broadly concordant with a genetic break between the Black Sea and the Mediterranean.

While the topic is of broad interest to the fields of phylogeography and population genetics with particular attention to admixture zones, I have several comments/clarifications that need to be addressed before the paper is acceptable for publication.

Major comments:

- Sample information. Suggest to include the information on Basin (Table S1) in Figure 1 perhaps as part of the Figure caption, for easier reference.

Basin information added to Figure 1 accordingly.

- l.247 Indicate reason for excluding individuals with STRUCTURE q<0.7 from pairwhise Phi-ST analysis?

We previously excluded those individuals so that we have individuals clearly belonging one or to the other cluster for this analysis. However, now we did not exclude any individuals and repeated the Fst analyses. 

- l.304. Indicate overall Fst value and associated p-value (microsatellites) to support statement of significant differentiation among sampling sites 

Thank you, global Fst value and its significance was added here.

- L.305. On the Dardanelles samples - indicate population number or code at the first instance for easy reference. The detail for the Dardanelle samples being population #10 was only mentioned later in line 370.

The sampling code for Dardanelles was added here. 

- Table 1. For correction, the table caption indicates "nuclear genetic divergence", but it includes Phi-ST values from mitochondrial data

Caption of Table 1 (now Fig 3) corrected accordingly. Now it also mentions “mitochondrial”.

- Table 1. Suggestions: (1) Indicate Basin or Group (Black Sea, TSS, Mediterannean etc...) on Table 1 for easier reference and (2) Might be more visually informative to make this into a heatmap (retaining the values for reference).

Thank you for this suggestion. We indicated 3 groups in table 1 (now Fig 3) (with different colors) and also made it into a heatmap.

- AMOVA. Table 2 and 3 can be combined in a single table.

Thank you. Tables 2 and 3 are now combined and named as “Table 2”. 

- l.350. H7 is not geographically restricted to the Black Sea, as it is found in the Mediterranean samples.

The sentence is corrected accordingly. 

- On genetic structure, it is important to state first whether the mitochondrial data reveal significant genetic structure (Phi ST > 0), before presenting the AMOVA results. Also, what is the overall Phi-ST value and its associated p-value? 

Thank you. This information is now added before mentioning AMOVA results.

(There was significant genetic differentiation in the overall data, with the overall PhiST value of 0.377 (p=0.0004). )

- l.385. The same two hypothesis of geographic structure tested with microsatellites

- AMOVA Tables 5 and 6 can be combined 

Thank you. Tables 5 and 6 are now combined and named as “Table 5”. 

On comparison between microsatellite and mt CoI results:

- On line 453, the authors report that microsatellites and mtDNA did not reflect the same degree of divergence in the Mediterranean. What does 'degree of divergence' mean? Are they referring to the Fst and Phi-st values? It will be difficult to compare divergence for different marker types. If anything, what is clear is that there is a broadly concordant pattern of geographic structure revealed by both markers. Marked genetic groups for the microsatellite data is clear (clusters M and C, with differential distributions across the sampled range). For the mtDNA data, yes the sequence divergence may be low (with closely related haplotypes separated by 1-2 bp only), however, their markedly different geographical distributions (H7 predominant in the Black Sea and TSS, H2 and H4 predominant in the Mediterranean and Atlantic) also give rise to a geographic break broadly consistent with the microsatellite data. This is also seen in the AMOVA for both markers (Tables 3, 4, 5 and 6).

- I suggest that the authors present the cytonuclear pattern (microsatellite cluster - mtDNA haplotype combination) to gain further insight into the patterns of admixture in the TSS. To what extent do individuals with belonging to different microsatellite clusters share mitochondrial haplotypes? 

We added a Supplementary Figure (S4 Fig) showing both main haplogroups of mtDNA and clusters of microsatellites. Here, it is seen that in the Black Sea and the TSS, individuals from Cluster C share the same mitochondrial haplogroup whereas in the sympatric clusters in the Mediterranean, haplogroups do not always correspond to microsat clusters. In other words, mtDNA CO1 haplogroups are observed in both STRUCTURE clusters in the Mediterranean.

- Suggest to perform isolation with migration (IMA) analyses to test hypotheses regarding mitochondrial gene flow between microsatellite clusters. it might be possible to test hypotheses regarding directionality of mitochondrial gene flow, as well as infer whether there was gene flow after divergence (or none).

We run IMA several times under different conditions. However, the results we were obtaining, although similar, were in the range of million years for the splitting time. As such they could not be interpreted. Probably using only one, and relatively short, marker could not provide enough information for the meaningful results.

Minor comments/corrections

- l.70 North Sea-Baltic Sea

This is also corrected.

- l.125 connectivity among n populations

This is now corrected.

- l. 154. ... in A1. Do you mean S1 Table?

Yes, it is now changed to S1 Table.

- l.265 "didn't" replace with "did not"

Corrected accordingly.

---

## [Decision Letter · Decision Letter 1]

9 Feb 2022

PONE-D-21-13064R1Mitonuclear genetic patterns of divergence in the marbled crab, <pachygrapsus marmoratus=""> (Fabricius, 1787) along the Turkish seas</pachygrapsus>PLOS ONE

Dear Dr. Çetin,

Thank you for submitting your manuscript to PLOS ONE. After careful consideration, we feel that it has merit but does not fully meet PLOS ONE’s publication criteria as it currently stands. Therefore, we invite you to submit a revised version of the manuscript that addresses the points raised during the review process.

The manuscript is greatly improved and the reviewers were generally pleased with the ammendments. Few minor things remain.

Main points.

1.    The higher sign in AMOVA with clusters derived from the data is ENTIRELY expected. the data is analyzed for structure, and then the data are analyzed with respect to that structure (circlular). I recommend you drop the Amova on clusters, and only keep the geographic regions and Sampling site anova up in the table. You can test in cross, variance in microsats within clusters defiend from mtDNA – and vice versa.

2.    Rev 2 asked for Samova analyses. I say more analyses are not needed (unless you want to give it a go). But you should discuss that doing Samova might be interesting (also on more markers) and the K3-5 heterogeneity shown by the structure analyses, and that possible explanations for these results. Are these non-biological (technical), multiple ancestral populations that mixed, or ecomorphs within species,…etc

3.    Check that all figures and tables including for the supplementary is refereed correctly as e.g., the S2 figure is not refereed to in the text. At the same time S1 Fig, S2 Table, and S3 Table was not included in the submission (rev 2).

Minor points.

New opening line of abstract is better, but the ending is a bit strange. Consider rewriting and switching verbs, “created” seems a bit off.

“because secondary contact zones of post-glacial lineages can be created.”

Line 84. Add reference

“These transition zones are often regions where divergent lineages meet in secondary contact.”

We look forward to receiving your revised manuscript.

Kind regards,

Arnar Palsson, Ph.D.

Academic Editor

PLOS ONE

Journal Requirements:

Additional Editor Comments:

The manuscript is greatly improved and the reviewers were generally pleased with the ammendments. Few minor things remain.

Main points.

1. The higher sign in AMOVA with clusters derived from the data is ENTIRELY expected. the data is analyzed for structure, and then the data are analyzed with respect to that structure (circlular). I recommend you drop the Amova on clusters, and only keep the geographic regions and Sampling site anova up in the table. You can test in cross, variance in microsats within clusters defiend from mtDNA – and vice versa.

2. Rev 2 asked for Samova analyses. I say more analyses are not needed (unless you want to give it a go). But you should discuss that doing Samova might be interesting (also on more markers) and the K3-5 heterogeneity shown by the structure analyses, and that possible explanations for these results. Are these non-biological (technical), multiple ancestral populations that mixed, or ecomorphs within species,…etc

3. Check that all figures and tables including for the supplementary is refereed correctly as e.g., the S2 figure is not refereed to in the text. At the same time S1 Fig, S2 Table, and S3 Table was not included in the submission (rev 2).

Minor points.

New opening line of abstract is better, but the ending is a bit strange. Consider rewriting and switching verbs, “created” seems a bit off.

“because secondary contact zones of post-glacial lineages can be created.”

Line 84. Add reference

“These transition zones are often regions where divergent lineages meet in secondary contact.”

Reviewers' comments:

Reviewer's Responses to Questions

**Comments to the Author**

1. If the authors have adequately addressed your comments raised in a previous round of review and you feel that this manuscript is now acceptable for publication, you may indicate that here to bypass the “Comments to the Author” section, enter your conflict of interest statement in the “Confidential to Editor” section, and submit your "Accept" recommendation.

Reviewer #1: All comments have been addressed

Reviewer #2: (No Response)

Reviewer #3: All comments have been addressed

2. Is the manuscript technically sound, and do the data support the conclusions?

Reviewer #1: Yes

Reviewer #2: Yes

Reviewer #3: Yes

3. Has the statistical analysis been performed appropriately and rigorously? 

Reviewer #1: Yes

Reviewer #2: Yes

Reviewer #3: Yes

4. Have the authors made all data underlying the findings in their manuscript fully available?

Reviewer #1: Yes

Reviewer #2: Yes

Reviewer #3: Yes

5. Is the manuscript presented in an intelligible fashion and written in standard English?

Reviewer #1: Yes

Reviewer #2: Yes

Reviewer #3: Yes

6. Review Comments to the Author

Reviewer #1: I’m happy with the amendments the authors have made and want to congratulate the authors on making it this far. As I’m happy with the amendments, I don’t have any more comments, except that the authors need to check that all figures and tables including for the supplementary is refereed correctly as e.g., the S2 figure is not refereed to in the text. At the same time S1 Fig, S2 Table, and S3 Table was not included in the submission. But even so I’m happy with the amendments.

Reviewer #2: General

I think the manuscript has been considerably improved and my major concerns have been mostly resolved. However, I still have one concern about the results of AMOVA.

Major comments:

Regarding comments by Reviewer 1 (32) and reply to my previous comment (45): I agree with Reviewer 1. The AMOVA results should be more carefully discussed. I think the AMOVA test showing more variation within groups than between groups implies that the grouping by geographical regions does not sufficiently represent population genetic structure.

I therefore recommend you to conduct SAMOVA with sampling site data, not by regions or clusters C/M, since SAMOVA can detect the most plausible grouping by comparing FCT values of different K without information for grouping in advance. I do not agree with your opinion that "SAMOVA won't add anything new to our interpretation."

Minor comments:

Line 44. SoM is not obvious in Abstract. Please spell out it for the first appearance.

Line 126. Mediterranean => the Mediterranean / Gbif => GBIF

Line 142. ", [27]" => " [27], "

Line 280–282. I think it is also important for Pm79 that numbers of 212 and 214 are almost same in red but 214 dominates in green.

Line 494-496. Explaining the similarity of COI sequences by its slow mutation rate may contradict your discussion in Line 438-440 that mtDNA sorts faster.

Reviewer #3: The authors have incorporated the comments and suggestions to the manuscript, with the exception of one suggested analysis (IMA), which they have provided justification for (results do not provide meaningful estimates of divergence time because of very wide variation). The manuscript is much improved incorporating all the other comments from the reviewers.

7. PLOS authors have the option to publish the peer review history of their article (what does this mean?). If published, this will include your full peer review and any attached files.

Reviewer #1: **Yes: **Charles Christian Riis Hansen

Reviewer #2: No

Reviewer #3: No

---

## [Author Response · Author response to Decision Letter 1]

21 Mar 2022

We thank the editor and the reviewers for their comments and for the opportunity to submit a revised draft. Below, we answer the specific comments and refer to changes we made.

Main points.

1. The higher sign in AMOVA with clusters derived from the data is ENTIRELY expected. the data is analyzed for structure, and then the data are analyzed with respect to that structure (circlular). I recommend you drop the Amova on clusters, and only keep the geographic regions and Sampling site anova up in the table. You can test in cross, variance in microsats within clusters defiend from mtDNA – and vice versa.

Yes, we agree that it is expected to have the same signal in AMOVA as STRUCTURE clusters for microsatellites. Thus, we removed that part from Table 1. However, we kept the part where we tested variance in mtDNA explained by microsatellite clusters in Table 3, as it is a cross-test. 

2. Rev 2 asked for Samova analyses. I say more analyses are not needed (unless you want to give it a go). But you should discuss that doing Samova might be interesting (also on more markers) and the K3-5 heterogeneity shown by the structure analyses, and that possible explanations for these results. Are these non-biological (technical), multiple ancestral populations that mixed, or ecomorphs within species,…etc

Thank you very much for your comment. We agree that new analyses are not needed. Since the K3-5 heterogeneity of the STRUCTURE analysis mainly involves cluster C with no geographical pattern (they are both present in the Black Sea and the Mediterranean), we believe this could be an artefact of the small number of markers used or a reflection of chaotic genetic patchiness. We also think further analyses with more markers will be needed and interesting to understand the detailed geographical pattern better. 

Now we also added a paragraph in our discussion about this:

Line 521-532: “Larger genetic variation within sampling sites than those between them (see Table 1 and S5 Fig) suggest a heterogenic genetic pool which was also observed in previous studies across the Italian [47–49], Tunisian [51], Portuguese [53] and Ligurian coasts [54]. Additionally, K=3 barplot of the STRUCTURE analysis showed heterogeneity of sampling sites mainly involving cluster C with no geographical structure (S2 Fig). Previously suggested ecological and biological processes such as differences in reproductive success, larval dispersion pattern and local larval retention could also be responsible for heterogeneity of the genetic pool observed in this study [54]. However, heterogeneity of STRUCTURE clusters at K 3-5 could also be an artefact of the small number of markers used. We think that by using more genetic markers such as SNPs, better inferences on the drivers of genetic structure in the region could be possible with more detailed analyses such as Spatial Analysis of Molecular Variance (SAMOVA) and genetic relatedness.”

3. Check that all figures and tables including for the supplementary is refereed correctly as e.g., the S2 figure is not refereed to in the text. At the same time S1 Fig, S2 Table, and S3 Table was not included in the submission (rev 2).

Thank you, now all of the supplementary material is included. S2 Figure is also referred to in the text:

Line 297: “Barplots for other K values (K: 3-5) were visually inspected (S2 Fig). At K=3, some individuals belonging to Cluster C formed a third cluster, but this did not correspond to any geographical pattern (S2 Fig).”

Minor points.

New opening line of abstract is better, but the ending is a bit strange. Consider rewriting and switching verbs, “created” seems a bit off.

“because secondary contact zones of post-glacial lineages can be created.”

Thank you, we now changed the verb to “formed” (Line 35).

Line 84. Add reference

“These transition zones are often regions where divergent lineages meet in secondary contact.”

A reference is now added here accordingly (Line 71).

6. Review Comments to the Author

Reviewer #1: I’m happy with the amendments the authors have made and want to congratulate the authors on making it this far. As I’m happy with the amendments, I don’t have any more comments, except that the authors need to check that all figures and tables including for the supplementary is refereed correctly as e.g., the S2 figure is not refereed to in the text. At the same time S1 Fig, S2 Table, and S3 Table was not included in the submission. But even so I’m happy with the amendments.

Reviewer #2: General

I think the manuscript has been considerably improved and my major concerns have been mostly resolved. However, I still have one concern about the results of AMOVA.

Major comments:

Regarding comments by Reviewer 1 (32) and reply to my previous comment (45): I agree with Reviewer 1. The AMOVA results should be more carefully discussed. I think the AMOVA test showing more variation within groups than between groups implies that the grouping by geographical regions does not sufficiently represent population genetic structure.

I therefore recommend you to conduct SAMOVA with sampling site data, not by regions or clusters C/M, since SAMOVA can detect the most plausible grouping by comparing FCT values of different K without information for grouping in advance. I do not agree with your opinion that "SAMOVA won't add anything new to our interpretation."

Thank you very much for your suggestion. In addition to our previous answer to Reviewer 1 (32), we think Fst table and Structure map would show the signal if SAMOVA would be picking up a previously undetected pattern. Additionally, PcoA analysis (S5 Fig) does not show any groupings even within the sampling sites. However, we believe that with more genetic markers (such as SNPs) and sampling effort, more detailed analyses such as SAMOVA could be useful for inferring the drivers of genetic structure in the region.

We now also discuss AMOVA in this perspective in the discussion:

Line 521-532: “Larger genetic variation within sampling sites than those between them (see Table 1 and S5 Fig) suggest a heterogenic genetic pool which was also observed in previous studies across the Italian [47–49], Tunisian [51], Portuguese [53] and Ligurian coasts [54]. Additionally, K=3 barplot of the STRUCTURE analysis showed heterogeneity of sampling sites mainly involving cluster C with no geographical structure (S2 Fig). Previously suggested ecological and biological processes such as differences in reproductive success, larval dispersion pattern and local larval retention could also be responsible for heterogeneity of genetic pool in this study [54]. However, heterogeneity of STRUCTURE clusters at K 3-5 could also be an artefact of the small number of markers used. We think that using more genetic markers such as SNPs, better inferences of the drivers of genetic structure in the region could be possible with more detailed analyses such as Spatial Analysis of Molecular Variance (SAMOVA) and genetic relatedness”

Minor comments:

Line 44. SoM is not obvious in Abstract. Please spell out it for the first appearance.

Thank you, now it is spelled out (Line 44).

Line 126. Mediterranean => the Mediterranean / Gbif => GBIF

Corrected accordingly.(Line 130)

Line 142. ", [27]" => " [27], "

Corrected accordingly (now it is the reference [30] instead of 27.). (Line 146)

Line 280–282. I think it is also important for Pm79 that numbers of 212 and 214 are almost same in red but 214 dominates in green.

We now added a sentence to the next paragraph about this: 

Line 310: “…Cluster C had a dominant allele for the locus Pm79 whereas M had almost equal frequencies of its alleles.”.

Line 494-496. Explaining the similarity of COI sequences by its slow mutation rate may contradict your discussion in Line 438-440 that mtDNA sorts faster.

Here, mtDNA sorting faster because of its smaller effective population size does not necessarily contradict our discussion. Because of this, we see a stronger effect of genetic drift in the Black Sea in the mtDNA. On the other hand, mutation rate of COI would be too slow for new haplotypes appearing and becoming prevalent in the last 20,000 years. As a result, it does not have enough resolution to help infer processes during the LGM, and thus could explain the similarity of haplotypes in the Mediterranean Sea.

Reviewer #3: The authors have incorporated the comments and suggestions to the manuscript, with the exception of one suggested analysis (IMA), which they have provided justification for (results do not provide meaningful estimates of divergence time because of very wide variation). The manuscript is much improved incorporating all the other comments from the reviewers.

Thank you very much.

---

## [Editor Report · Decision Letter 2]

23 Mar 2022

Mitonuclear genetic patterns of divergence in the marbled crab, <pachygrapsus marmoratus=""> (Fabricius, 1787) along the Turkish seas

PONE-D-21-13064R2</pachygrapsus>

Dear Dr. Çetin,

We’re pleased to inform you that your manuscript has been judged scientifically suitable for publication and will be formally accepted for publication once it meets all outstanding technical requirements.

Kind regards,

Arnar Palsson, Ph.D.

Academic Editor

PLOS ONE
---

## [Editor Report · Acceptance letter]

25 Mar 2022

PONE-D-21-13064R2 

Mitonuclear genetic patterns of divergence in the marbled crab, *Pachygrapsus marmoratus* (Fabricius, 1787) along the Turkish seas 

Dear Dr. Çetin:

I'm pleased to inform you that your manuscript has been deemed suitable for publication in PLOS ONE. Congratulations! Your manuscript is now with our production department. 

Kind regards, 

on behalf of

Dr. Arnar Palsson 

Academic Editor

PLOS ONE